# Neural Tangent Kernel Eigenvalues Accurately Predict Generalization

## Abstract

In many cases, infinitely-wide deep neural networks are equivalent to kernel regression using the network's "neural tangent kernel." With the aim of shedding light on the learning behavior of wide neural networks, in this work we derive a new theory predicting the generalization of kernel regression. Our theory accurately predicts not only test mean-squared-error but all first- and second-order statistics of the learned function. Furthermore, using a measure quantifying the "learnability" of a given target function, we prove a new "no-free-lunch" theorem characterizing a fundamental tradeoff in the inductive bias of kernel regression: improving generalization for a given target function must worsen its generalization for orthogonal functions. We further demonstrate the utility of our theory by analytically predicting two surprising phenomena — worse-than-chance generalization on hard-to-learn functions and nonmonotonic error curves in the small data regime — which we subsequently observe in experiments. Though our theory is derived for networks in the infinite-width limit, we experimentally find that its predictions are accurate for wide finite networks.

## 1 Introduction

Understanding and predicting a machine learning model's generalization to unseen data is a central goal of machine learning theory. For a given class of model, one would ideally want a simple picture of a model's inductive bias, identifying the set of functions on which a given model will generalize well and those on which it will generalize poorly and making quantitative predictions of key measures of the quality of generalization. In this paper, we derive such a theory for ridgeless kernel regression with any kernel and, using the neural tangent kernel (NTK) equivalence between kernel regression and infinitely wide deep neural networks, shed light on the inductive bias and generalization of wide deep neural networks.

Our main contributions are as follows:

- We prove a basic conservation law describing the inductive bias of kernel regression (Theorem 1). This law states that any kernel has a fixed budget of a quantity we call "learnability" that it must allocate to an orthogonal basis of functions, and this budget is equal to the size of the training set. As a consequence of this conservation law, we prove that for every kernel, there exists a target function on which kernel regression generalizes worse than chance (Corollary 1), and we provide a recipe to construct such functions.

- We derive a new theory of generalization for kernel regression, culminating in analytical expressions for all first- and second-order statistics of the learned function (Equations 11 and 13). This theory extends the spectral picture of kernel regression revealed by Bordelon et al. (2020) and Canatar et al. (2021). We conclude from our theory that, in realistic settings, most kernel eigenfunctions have the counterintuitive property that MSE *increases* as examples are added to a small training set (Section 2.7).

- We empirically verify all our results on synthetic datasets using both exact kernel regression and deep networks of width 500 trained with gradient descent. We find that our conservation law and analytical expressions for generalization performance hold to an excellent approximation even for wide finite networks, and we observe worse-than-chance generalization and increasing MSE as predicted. We find that our theory's core predictions remain

fairly accurate even down to width 20 for depth-four networks, suggesting it is a promising starting point for understanding generalization in practical architectures.

## 1.1 RELATED WORK

The study of the generalization of kernel regression via spectral analysis began in the literature on Gaussian process inference, for which kernel regression gives the mean of the posterior function. In a limited teacher-student setting in which teacher and student were described by the same Gaussian process, Sollich (1999) and Vivarelli & Opper (1999) studied expected MSE as a function of training set size, and Sollich (2001) extended their results to the setting in which the Gaussian processes' eigenvalues can differ, but no similar results from this era described the fully general setting.

The discovery of the equivalence between wide neural networks and kernel regression via the NTK (Jacot et al., 2018; Lee et al., 2019) sparked a resurgence of interest in the generalization of kernel regression (Belkin et al., 2018a; 2019b;a; Liang & Rakhlin, 2020; Bietti & Mairal, 2019). Further noting that both kernel regression and deep learning can generalize well despite perfectly interpolating their training data, Belkin et al. (2018b) argued that "to understand deep learning we need to understand kernel learning." Arora et al. (2019a) derived a data-dependent generalization bound for a wide two-layer architecture involving its infinite-width NTK. Though this bound is a significant advance over VC-dimension-based bounds inapplicable to overparameterized models, it only applies to a specific architecture and can be many times greater than the true loss, while our results predict true loss within small error bars and apply to any infinite-width architecture (and in fact to any incarnation of kernel regression).

In pioneering work, Bordelon et al. (2020) and Canatar et al. (2021) extended classic results on the generalization of kernel regression to the fully general case and, using the NTK, confirmed that their approximate expressions for expected MSE agree even with wide *finite* deep neural networks. Their results reveal a simple picture of neural network generalization: as samples are added to the training set, the network generalizes well on a larger and larger subspace of input functions. The natural basis for this subspace of learnable functions is the eigenbasis of the NTK, and its eigenfunctions are learned in descending order of their eigenvalues. The results of the present paper corroborate and extend this picture of the generalization of kernel regression. Our work chiefly differs from theirs in that (a) our conservation law is new, (b) we derive and test expressions for generic first- and second-order statistics of the learned function, not just MSE, and (c) our derivation is quite different (and, we believe, easier to understand) even when it arrives at the same results.

Our work is related to the well-known observation that neural networks have a "spectral bias" towards representing slowly-varying functions (Valle-Perez et al., 2018; Yang & Salman, 2019); in particular, the kernel spectrum picture clarifies that this bias is a consequence of the fact that high NTK eigenmodes are typically slowly-varying. We also note a body of work studying the related phenomenon that slowly-varying functions are learned first during training (Rahaman et al., 2019; Xu et al., 2019b;a; Xu, 2018; Cao et al., 2019; Su & Yang, 2019).

## 2 THEORY

### 2.1 A REVIEW OF KERNEL REGRESSION

Consider the task of learning an $m$-element function $f : \mathcal{X} \to \mathbb{R}^m$ given a set of $n$ unique training points $\mathcal{D} = \{x_i\}_{i=1}^n \subseteq \mathcal{X}$ and their corresponding function values $f(\mathcal{D}) \in \mathbb{R}^{n \times m}$. To simplify our analysis, we will let the domain $\mathcal{X}$ be discrete with size $M \equiv |\mathcal{X}|$ and assume the $n$ training points are uniformly sampled from $\mathcal{X}$. This choice of discrete domain will permit us to use matrices and vectors instead of operators and functions in our derivations. By taking $M \to \infty$, our results easily extend to problems with continuous domain: for example, if $M \to \infty$ and we allow the points in $\mathcal{X}$ to approach a density equal to some desired measure over $\mathbb{R}^d$, we recover the standard setting where the data are sampled nonuniformly from $\mathbb{R}^d$.

We will use $\hat{f}$ to denote the function learned by a neural network trained on this dataset. Remarkably, for an infinite-width neural network optimized via gradient descent to zero training MSE loss, this learned function is given by

$$\hat{f}(x) = K(x, \mathcal{D}) K(\mathcal{D}, \mathcal{D})^{-1} f(\mathcal{D}), \qquad (1)$$

where $K : \mathcal{X} \times \mathcal{X} \to \mathbb{R}$ is the network's "neural tangent kernel" (NTK) (Jacot et al., 2018; Lee et al., 2019), $K(\mathcal{D}, \mathcal{D})$ is the "kernel matrix" defined by $K(\mathcal{D}, \mathcal{D})_{ij} = K(x_i, x_j)$, and $K(x, \mathcal{D})$ is a row vector with components $K(x, \mathcal{D})_i = K(x, x_i)$.[1] We give a brief introduction to the NTK in Appendix C. Due to its similarity to the normal equation of linear regression, Equation 1 is often called "kernel regression."[2]

Equation 1 holds exactly in the infinite-width limit of fully-connected networks (Lee et al., 2019), convolutional networks (Arora et al., 2019b), transformers (Hron et al., 2020), and more (Yang, 2019). Moreover, several empirical studies have shown it to be a good approximation for networks of even modest width (Lee et al., 2019; 2020). Our approach will be to study the generalization behavior of Equation 1, conjecture that our results also apply to finite networks, and finally provide strong support for our conjecture with experiments.

Examining Equation 1, one finds that the $m$ indices of $f$ can each be treated separately: the learned $\hat{f}$ is equivalent to simply performing kernel regression with each of the $m$ indices as a scalar target function and then vectorizing the results. For simplicity, then, we hereafter assume $m = 1$. The extension to $m > 1$ is straightforward.

## 2.2 FIGURES OF MERIT OF $\hat{f}$

We will study three measures of the quality of the learned function $\hat{f}$. All three will be defined in terms of the inner product over $\mathcal{X}$: for two functions $g, h : \mathcal{X} \to \mathbb{R}$, their inner product is $\langle g, h \rangle \equiv \frac{1}{M} \sum_{x \in \mathcal{X}} g(x)h(x)$.

The first measure of quality is mean-squared error (MSE). For a particular dataset $\mathcal{D}$, MSE is given by $\mathcal{E}^{(\mathcal{D})}(f) \equiv \langle f - \hat{f}, f - \hat{f} \rangle$. Of more interest will be the expected MSE over all datasets of size $n$, given by $\mathcal{E}(f) \equiv \mathbb{E}_{\mathcal{D}}\big[\mathcal{E}^{(\mathcal{D})}(f)\big]$. We note that the inner product is taken over all $\mathcal{X}$, including $\mathcal{D}$, even though $\hat{f}(x) = f(x)$ for $x \in \mathcal{D}$ for kernel regression.

In maximizing the similarity of $\hat{f}$ to $f$, we typically wish to minimize its similarity to all functions orthogonal to $f$. The second measure examines the coefficient in $\hat{f}$ of one such orthogonal function to $f$. Letting $g : \mathcal{X} \to \mathbb{R}$ be a function such that $\langle f, g \rangle = 0$, we consider the mean and variance of the quantity $\langle \hat{f}, g \rangle$. We will derive accurate predictions for this metric of generalization.

Lastly, we introduce a figure of merit quantifying the alignment of $f$ and $\hat{f}$, which we call "learnability." It is given by

$$\mathcal{L}^{(\mathcal{D})}(f) \equiv \frac{\langle f, \hat{f} \rangle}{\langle f, f \rangle}, \quad \mathcal{L}(f) \equiv \mathbb{E}_{\mathcal{D}}\Big[\mathcal{L}^{(\mathcal{D})}(f)\Big], \tag{2}$$

where $\mathcal{L}^{(\mathcal{D})}(f)$ is the dataset-dependent learnability of the function $f$ ("$\mathcal{D}$-learnability") and $\mathcal{L}(f)$ is its expectation over random data ("learnability"). Though at first glance these two seem like odd figures of merit, we will soon show that they have many desirable properties when $\hat{f}$ is given by Equation 1: unlike MSE, both are bounded in $[0, 1]$ (Lemma 1d), always change monotonically as new data points are added (Lemma 1e), are invariant to rescalings of $f$, and obey a simple conservation law (Theorem 1). Furthermore, expanding the inner product in the definition of $\mathcal{E}$ and noting that $\mathcal{E}(f) \geq \langle f, f \rangle (1 - \mathcal{L}(f))^2$, one can see that low MSE is impossible without high learnability. We will ultimately derive an accurate approximation for learnability that is substantially simpler than any known approximation for MSE.

---

[1] Naively, Equation 1 is only the *expected* learned function, and the true learned function will include a fluctuation term reflecting the random initialization. However, by storing a copy of the parameters at $t = 0$ and redefining $\hat{f}_t(x) := \hat{f}_t(x) - \hat{f}_0(x)$ throughout optimization and at test time, this term becomes zero, and so we neglect it in our theory and use this trick in our experiments.

[2] Interestingly, exact Bayesian inference for infinite-width neural networks yields predictions of the same form as Equation 1, with $K$ being the "neural network Gaussian process" (NNGP) kernel instead of the NTK (Lee et al., 2018). We will proceed treating $K$ as a network's NTK, but our theory and exact results (including our "no-free-lunch" theorem) apply equally well to any incarnation of kernel regression, including RBF kernel regression and linear regression.

## 2.3 THE KERNEL EIGENSYSTEM

By definition, any kernel function is symmetric and positive-semidefinite (Shawe-Taylor et al., 2004). This implies that we can find a set of orthonormal eigenfunctions $\{\phi_i\}_{i=1}^M$ and nonnegative eigenvalues $\{\lambda_i\}_{i=1}^M$ that satisfy

$$\frac{1}{M} \sum_{x' \in \mathcal{X}} K(x, x') \phi_i(x') = \lambda_i \phi_i(x), \quad \langle \phi_i, \phi_j \rangle = \delta_{ij}. \tag{3}$$

For simplicity (and to ensure that $K(\mathcal{D}, \mathcal{D})$ is invertible), we will assume that $K$ is in fact positive *definite* and $\lambda_i > 0$, an assumption that will hold in most cases of interest.[3,4]

We will now translate Equation 1 to this eigenbasis. First we decompose $f$ and $\hat{f}$ into weighted sums of the eigenfunctions as

$$f(x) = \sum_{i=1}^M v_i \phi_i(x), \quad \hat{f}(x) = \sum_{i=1}^M \hat{v}_i \phi_i(x), \tag{4}$$

where $\mathbf{v}$ and $\hat{\mathbf{v}}$ are vectors of coefficients. Using this notation, MSE is $\mathcal{E}^{(\mathcal{D})} = (\mathbf{v} - \hat{\mathbf{v}})^2$ and $\mathcal{D}$-learnability is $\mathcal{L}^{(\mathcal{D})} = \mathbf{v}^T \hat{\mathbf{v}} / |\mathbf{v}|^2$.

Noting that $K(x_1, x_2) = \sum_{i=1}^M \lambda_i \phi_i(x_1) \phi_i(x_2)$, we can decompose the kernel matrix as $K(\mathcal{D}, \mathcal{D}) = \mathbf{\Phi}^T(\mathcal{D}) \mathbf{\Lambda} \mathbf{\Phi}(\mathcal{D})$, where $\mathbf{\Phi}(\mathcal{D})_{ij} = \phi_i(x_j)$ is the $M \times n$ "design matrix" and $\mathbf{\Lambda} = \mathrm{diag}(\lambda_1, ..., \lambda_M)$ is a diagonal matrix of eigenvalues. The learned coefficients $\hat{\mathbf{v}}$ are then given by $\hat{v}_i = \langle \phi_i, \hat{f} \rangle = \lambda_i \phi_i(\mathcal{D}) K(\mathcal{D}, \mathcal{D})^{-1} \mathbf{\Phi}^T(\mathcal{D}) \mathbf{v}$. Stacking these coefficients into a matrix equation, we find that

$$\hat{\mathbf{v}} = \mathbf{\Lambda} \mathbf{\Phi}(\mathcal{D}) \left( \mathbf{\Phi}^T(\mathcal{D}) \mathbf{\Lambda} \mathbf{\Phi}(\mathcal{D}) \right)^{-1} \mathbf{\Phi}^T(\mathcal{D}) \mathbf{v} = \mathbf{T}^{(\mathcal{D})} \mathbf{v}, \tag{5}$$

where $\mathbf{T}^{(\mathcal{D})} \equiv \mathbf{\Lambda} \mathbf{\Phi}(\mathcal{D}) \left( \mathbf{\Phi}^T(\mathcal{D}) \mathbf{\Lambda} \mathbf{\Phi}(\mathcal{D}) \right)^{-1} \mathbf{\Phi}^T(\mathcal{D})$ is an $M \times M$ matrix, independent of $f$, that fully describes the model's learning behavior on a training set $\mathcal{D}$. We call this fundamental quantity the "learning transfer matrix." If we can determine the statistical properties of this matrix, we will understand the learning behavior of our model.

We also define the mean learning transfer matrix $\mathbf{T} \equiv \mathbb{E}_{\mathcal{D}} \left[ \mathbf{T}^{(\mathcal{D})} \right]$. $\mathcal{D}$-learnability and learnability are then respectively given by $\mathcal{L}^{(\mathcal{D})}(f) = \mathbf{v}^T \mathbf{T}^{(\mathcal{D})} \mathbf{v} / |\mathbf{v}|^2$ and $\mathcal{L}(f) = \mathbf{v}^T \mathbf{T} \mathbf{v} / |\mathbf{v}|^2$.

## 2.4 EXACT RESULTS

The following lemma gives basic properties of the quantities defined above.

**Lemma 1.** *The following properties of* $\mathbf{T}^{(\mathcal{D})}$, $\mathbf{T}$, $\mathcal{L}^{(\mathcal{D})}$, *and* $L$ *hold:*

(a) $\mathcal{L}^{(\mathcal{D})}(\phi_i) = \mathbf{T}_{ii}^{(\mathcal{D})}$, *and* $\mathcal{L}(\phi_i) = \mathbf{T}_{ii}$.

(b) *When* $n = 0$, $\mathbf{T} = \mathbf{T}^{(\mathcal{D})} = 0$ *and* $\mathcal{L}(f) = \mathcal{L}^{(\mathcal{D})}(f) = 0$.

(c) *When* $n = M$, $\mathbf{T} = \mathbf{T}^{(\mathcal{D})} = \mathbf{I}_M$ *and* $\mathcal{L}(f) = \mathcal{L}^{(\mathcal{D})}(f) = 1$.

(d) *All eigenvalues of* $\mathbf{T}^{(\mathcal{D})}$ *are in* $\{0, 1\}$, *all eigenvalues of* $\mathbf{T}$ *are in* $[0, 1]$, *and so* $\mathcal{L}(f), \mathcal{L}^{(\mathcal{D})}(f) \in [0, 1]$.

(e) *Let* $\mathcal{D}_+$ *be* $\mathcal{D} \cup x$, *where* $x \in X, x \notin \mathcal{D}$ *is a new data point. Then* $\mathcal{L}^{(\mathcal{D}+)}(f) \geq \mathcal{L}^{(\mathcal{D})}(f)$.

(f) *For any* $i \in \{1, ..., M\}$, $\frac{d}{d\lambda_i} \mathbf{T}_{ii}^{(\mathcal{D})} \geq 0$, *hence* $\frac{d}{d\lambda_i} \mathcal{L}^{(\mathcal{D})}(\phi_i) \geq 0$.

(g) *For any* $i, j \in \{1, ..., M\}$, $i \neq j$, $\frac{d}{d\lambda_i} \mathbf{T}_{jj}^{(\mathcal{D})} \leq 0$, *hence* $\frac{d}{d\lambda_i} \mathcal{L}^{(\mathcal{D})}(\phi_j) \leq 0$.

---

[3]By Mercer's Theorem, one can also find the eigensystem of any kernel on a continuous input space $\mathcal{X}$.

[4]We do not use the Reproducing kernel Hilbert space (RKHS) formalism in this work, but we note that, by the Moore–Aronszajn theorem, the kernel $K$ defines a unique RKHS.

Property (a) in Lemma 1 formalizes the relationship between the transfer matrix and learnability. Properties (b-e) together give an intuitive picture of the learning process: the learning transfer matrix monotonically interpolates between zero and $\mathbf{I}_M$ as the training set grows — adding data never harms the learnability of any function. Properties (f-g) show that the kernel eigenmodes are in competition: increasing one eigenvalue can only improve the learnability of the corresponding eigenfunction, but can only decrease the learnabilities of all others. We prove Lemma 1 in Appendix D.

We now present our first major result.

**Theorem 1** ("No-free-lunch" theorem for kernel regression). *For any complete basis of orthogonal functions $\mathcal{F}$,*

$$\sum_{f \in \mathcal{F}} \mathcal{L}(f) = \sum_{f \in \mathcal{F}} \mathcal{L}^{(\mathcal{D})}(f) = n. \tag{6}$$

The proof, which hinges on the intermediate result that $\mathrm{Tr}\left[\mathbf{T}^{(\mathcal{D})}\right] = n$, is given in Appendix E. We note that this result is stronger than the ordinary "no-free-lunch" theorem for learning algorithms, which requires averaging over *all* target functions instead of merely an orthogonal basis (Wolpert, 1996). To understand the significance of this result, consider that one might naively hope to design a neural kernel that achieves generally high performance for all target functions $f$. Theorem 1 states that this is impossible: averaged over a complete basis of functions, *all kernels achieve the same learnability*. This exact result implies that, because there exist no universally high-performing kernels, *we must instead aim to choose a kernel whose high-eigenvalue modes align well with the function to be learned*. To our knowledge, this is the first exact result quantifying such a tradeoff in kernel regression or deep learning.

Illustrating a consequence of this tradeoff, the following theorem states that, for any kernel, there will always be functions poorly-aligned with the kernel's inductive bias on which the model generalizes as bad or worse than it would by simply predicting zero on all unseen test points.

**Corollary 1** (Negative generalization). *There is always at least one eigenfunction $\phi_i$ for which $\mathcal{L}(\phi_i) \leq \mathcal{L}_{naive}(\phi_i)$ and $E(\phi_i) \geq \mathcal{E}_{naive}(\phi_i)$, where $\mathcal{L}_{naive}$ and $\mathcal{E}_{naive}$ are the learnability and mean MSE given by a naive, non-generalizing model with predictions given by*

$$\hat{f}_{naive}(x) = \begin{cases} f(x) & x \in \mathcal{D} \\ 0 & x \notin \mathcal{D} \end{cases}. \tag{7}$$

This theorem follows from Theorem 1. We give a full proof in Appendix E. This negative generalization property will hold for any function with learnability less than or equal to $n/M$, including any weighted sum of eigenfunctions with this property.

### 2.5 DERIVING A CLOSED-FORM EXPRESSION FOR $\mathbf{T}$

We will now derive an approximation for $\mathbf{T}$ and the second moments of $\mathbf{T}^{(\mathcal{D})}$, ultimately yielding simple yet accurate expressions for $\mathcal{L}$, $\mathcal{E}$, and $\langle \hat{f}, g \rangle$. We sketch our method and state our results here and provide a full derivation in Appendix F.

We begin by noting that the expectation in $\mathbf{T} \equiv \mathbb{E}_{\mathcal{D}}\left[\mathbf{T}^{(\mathcal{D})}\right]$ is essentially an expectation over a combinatorially large set of possible design matrices $\mathbf{\Phi}(\mathcal{D})$, each of which has orthogonal columns. We replace this with an average over *all* $M \times n$ matrices $\mathbf{\Phi}$ satisfying $\mathbf{\Phi}^T \mathbf{\Phi} = M \mathbf{1}_n$, which we write

$$\mathbf{T} = \mathbb{E}_{\mathcal{D}}\left[\mathbf{\Lambda}\mathbf{\Phi}(\mathcal{D})\left(\mathbf{\Phi}^T(\mathcal{D})\mathbf{\Lambda}\mathbf{\Phi}(\mathcal{D})\right)^{-1}\mathbf{\Phi}^T(\mathcal{D})\right] \approx \mathbb{E}_{\mathbf{\Phi}}\left[\mathbf{\Lambda}\mathbf{\Phi}\left(\mathbf{\Phi}^T\mathbf{\Lambda}\mathbf{\Phi}\right)^{-1}\mathbf{\Phi}^T\right]. \tag{8}$$

We can then readily prove by symmetry that the off-diagonal elements of $\mathbf{T}$ vanish, leaving the question of how each diagonal element depends on the set of eigenvalues. To probe this, we consider adding an $(M+1)$-th "test eigenmode" to the problem, augmenting the principal quantities as

$$\mathbf{\Phi}^+ = \begin{bmatrix} \mathbf{\Phi} \\ \phi^T \end{bmatrix}, \quad \mathbf{\Lambda}^+ = \begin{bmatrix} \mathbf{\Lambda} & 0 \\ 0 & \lambda \end{bmatrix}, \quad \mathbf{T}^+ = \mathbb{E}_{\mathbf{\Phi}, \phi}\left[\mathbf{\Lambda}^+\mathbf{\Phi}^+\left(\mathbf{\Phi}^{+T}\mathbf{\Lambda}^+\mathbf{\Phi}^+\right)^{-1}\mathbf{\Phi}^{+T}\right]. \tag{9}$$

Using the Sherman-Morrison formula, we find that the new mode's learnability is given by

$$\mathbf{T}^+_{(M+1)(M+1)} = \mathbb{E}_{\mathbf{\Phi},\phi}\left[\frac{\lambda}{\lambda + \left[\phi^T\left(\mathbf{\Phi}^T\mathbf{\Lambda}\mathbf{\Phi}\right)^{-1}\phi\right]^{-1}}\right] = \mathbb{E}_{\mathbf{\Phi},\phi}\left[\frac{\lambda}{\lambda + C^{(\mathbf{\Phi},\phi)}}\right], \quad (10)$$

where $C^{(\mathbf{\Phi},\phi)} \equiv \left[\phi^T\left(\mathbf{\Phi}^T\mathbf{\Lambda}\mathbf{\Phi}\right)^{-1}\phi\right]^{-1}$ is a nonnegative dataset-dependent constant. For large $n$ and realistic eigenspectra, $C^{(\mathbf{\Phi},\phi)}$ is distributed tightly around its mean, and so we can replace it with a constant, which we call $C$. We emphasize that $C$ is the same for every eigenmode. Using Theorem 1 to obtain a constraint on $C$, we reach our ultimate approximation that

$$\mathbf{T}_{ij} = \delta_{ij}\frac{\lambda_i}{\lambda_i + C}, \quad \text{where } C \geq 0 \text{ satisfies } \sum_{i=1}^{M}\frac{\lambda_i}{\lambda_i + C} = n. \quad (11)$$

Using Lemma 1, we obtain the major result that

$$\mathcal{L}(f) = \frac{1}{|\mathbf{v}|^2}\sum_{i=1}^{M}\frac{\lambda_i}{\lambda_i + C}\mathbf{v}_i^2, \quad (12)$$

with $C$ given by Equation 11.

A function is thus more learnable the more weight it places in high eigenvalue modes. We show in Section 3 that Equation 12 is in excellent agreement with experiments using both kernel regression and trained finite networks.

The learnability of each eigenmode depends critically on the value of $C$. We now present several properties characterizing how $C$ depends on $n$.

**Lemma 2.** *For $C$ satisfying the constraint in Equation 11, with $\{\lambda_i\}_{i=1}^{M}$ ordered from greatest to least, the following properties hold:*

*(a) $C = \infty$ when $n = 0$, and $C = 0$ when $n = M$.*

*(b) $C$ is strictly decreasing with $n$.*

*(c) $C \leq \frac{1}{n-\ell}\sum_{i=\ell+1}^{M}\lambda_i$ for all $\ell \in \{0, ..., n-1\}$.*

*(d) $C \geq \lambda_\ell\left(\frac{\ell}{n} - 1\right)$ for all $\ell \in \{n, ..., M\}$.*

Properties (a-b), paired with Equation 11, paint a surprisingly simple picture of the learning process: as the training set grows, $C$ gradually decreases, and eigenmodes are learned high-to-low as $C$ passes each eigenvalue. Properties (c-d) provide bounds on $C$, which, in addition to then yielding bounds on learnability, can be used to furnish an initial guess when numerically solving for $C$.

## 2.6 SECOND-ORDER STATISTICS OF $\mathbf{T}$

We now have an expression for the mean of $\mathbf{T}^{(\mathcal{D})}$, but many quantities of interest, including MSE, depend on second-order fluctuations about that mean. In Appendix G, we show how, by taking a derivative with respect to $\mathbf{\Lambda}$, we can obtain expressions for the second-order statistics of $\mathbf{T}^{(\mathcal{D})}$ and, as a result, for MSE. Our main second-order result is that

$$\text{Cov}\left[\mathbf{T}_{ij}^{(\mathcal{D})}, \mathbf{T}_{k\ell}^{(\mathcal{D})}\right] = \frac{C\lambda_i\lambda_k\left(\delta_{ik}\delta_{j\ell} + \delta_{i\ell}\delta_{jk} - \delta_{ij}\delta_{k\ell}\right)}{q(\lambda_i + C)(\lambda_j + C)(\lambda_k + C)(\lambda_\ell + C)}, \quad \text{where } q \equiv \sum_{m=1}^{M}\frac{\lambda_m}{(\lambda_m + C)^2}. \quad (13)$$

Using Equation 13, we can study the admixtures of particular spurious modes in the learned function $\hat{f}$. For example, if $f = \phi_i$ and $i \neq j$, then we find that

$$\mathbb{E}\left[\langle\hat{f}, \phi_j\rangle\right] = 0, \quad \mathbb{E}\left[\langle\hat{f}, \phi_j\rangle^2\right] = \text{Var}\left[\mathbf{T}_{ji}^{(\mathcal{D})}\right] = \frac{C\lambda_j^2}{q(\lambda_i + C)^2(\lambda_j + C)^2}. \quad (14)$$

We note that $\mathbb{E}\left[\langle \hat{f}, \phi_j \rangle^2\right]$ increases with $\lambda_j$, reinforcing our broad conclusion that the model is biased towards high-eigenvalue modes. Finally, by noting that $\mathcal{E}(f) = \mathbb{E}_{\mathcal{D}}\left[|(\mathbf{T}^{(\mathcal{D})} - \mathbf{I}_M)\mathbf{v}|^2\right]$, we can recover the expression of Bordelon et al. (2020) for MSE:

$$\mathcal{E}(f) = \mathbf{v}^T \mathbf{E} \mathbf{v}, \quad \text{where} \quad \mathbf{E}_{ij} \equiv \delta_{ij} \frac{nC}{q} \frac{1}{(\lambda_i + C)^2}. \tag{15}$$

Our experiments corroborate existing evidence that this is an excellent approximation for MSE.

### 2.7 NONMONOTONIC MSE CURVES

Expanding $\mathbf{E}$ for small $n$, we find that

$$\mathbf{E}_{ii}\Big|_{n=0} = 1, \qquad \frac{d\mathbf{E}_{ii}}{dn}\Big|_{n=0} = \frac{\sum_j \lambda_j^2}{\left(\sum_j \lambda_j\right)^2} - \frac{2\lambda_i}{\sum_j \lambda_j}. \tag{16}$$

The second equation implies that $\frac{d\mathbf{E}_{ii}}{dn}\big|_{n=0} > 0$ for all modes $i$ such that $\lambda_i < (\sum_j \lambda_j^2)/(2\sum_j \lambda_j)$, which suggests that, for such low-eigenvalue modes, MSE *increases* as the training set size grows from zero. In practice, such low modes are commonplace; in fact, for a bounded kernel on a continuous input space (for which $M \to \infty$), an infinite number of arbitrarily low modes is inevitable due to the constraint that $\sum_i \lambda_i$ is bounded[5]. We therefore expect that there quite often exist functions for which a given neural network yields an MSE nonmonotonic with $n$. Figure A1 shows that experiments confirm this surprising first-principles prediction. We emphasize that this is a different, more generic phenomenon than that noted by Canatar et al. (2021), which required that $f$ was either noisy or placed weight in unlearnable zero-eigenvalue modes.

## 3 EXPERIMENTS

Here we describe experiments confirming our theoretical predictions for exact NTK regression and wide neural networks. Unless otherwise stated, all experiments used a fully-connected (FC) four-hidden-layer (4L) ReLU architecture with width 500. Because this model is FC, it has a rotation-invariant NTK (Lee et al., 2019). These experiments used three distinct input spaces $\mathcal{X}$. For each, the eigenmodes of $\mathcal{X}$ can be grouped into degenerate subsets indexed by $k \in \mathbb{N}$, with higher $k$ corresponding to faster variation in space. In all cases, we find that as $k$ increases, eigenvalues decrease, in concordance with the widespread belief that neural nets have a "spectral bias" towards slowly-varying functions (Rahaman et al., 2019; Canatar et al., 2021; Cao et al., 2019). We now describe these three input spaces. For full experimental details, see Appendix H.

**Discretized Unit Circle.** The simplest input space we consider is the discretization of the unit circle into $M$ points, $\mathcal{X} = \{(\cos(2\pi j/M), \sin(2\pi j/M)\}_{j=1}^{M}$. The eigenfunctions on this domain are $\phi_0(\theta) = 1$, $\phi_k(\theta) = \sqrt{2}\cos(k\theta)$, and $\phi_k'(\theta) = \sqrt{2}\sin(k\theta)$, for $k \geq 1$.

**Hypercube.** The next input space we consider is the set of vertices of the $d$-dimensional hypercube, $\mathcal{X} = \{-1, 1\}^d$, giving $M = 2^d$. The eigenfunctions on this domain are the subset-parity functions $\phi_s(x) = (-1)^{s^T x}$, where $s \in \{0, 1\}^d$ is a vector indicating the elements of $x$ to which the output is sensitive (Yang & Salman, 2019). Here we define $k = \sum_i s_i$.

**Hypersphere.** To illustrate that our results extend easily to continuous input spaces, we consider the $d$-sphere $\mathbb{S}^d = \{x \in \mathbb{R}^{d+1} | x^2 = d + 1\}$. The eigenfunctions on this domain are the hyperspherical harmonics (see, e.g., Frye & Efthimiou (2012); Bordelon et al. (2020)) which group into degenerate sets indexed by $k \in \mathbb{N}$. The corresponding eigenvalues decrease exponentially with $k$, and so when summing over all eigenmodes to compute $C$ and $q$, we simply truncate the sum at $k_{max} = 70$.

Figure I5 shows the 4L ReLU NTK eigenvalues on each domain, all of which decrease almost monotonically with increasing $k$. In particular, the eigenvalues on the unit circle roughly follow a power-law decay like the NTK spectra of common image datasets (e.g. Lee et al. (2020)). We now describe our experiments.

---

[5]This constraint follows from the fact that $\sum_i \lambda_i = \frac{1}{M} \sum_{x \in \mathcal{X}} K(x, x) < \infty$.

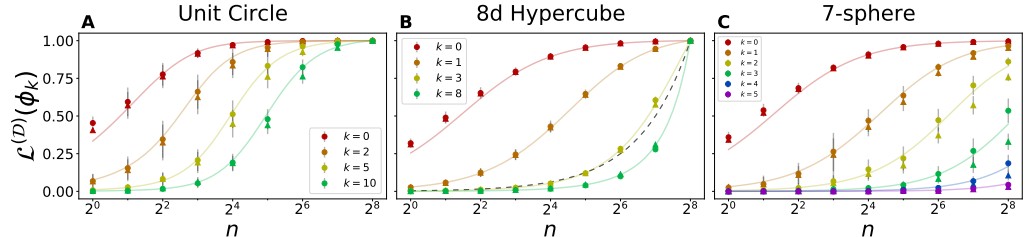

Figure 1: **Theoretical predictions closely match the true learnabilities of arbitrary eigenfunctions on diverse input spaces.** Each plot shows learnability $\mathcal{L}(\phi_k)$ (Equation 11) of several eigenfunctions as a function of training set size $n$. Theoretical curves show excellent agreement with results from exact NTK regression (triangles) and finite nets trained via gradient descent (circles). Error bars reflect $1\sigma$ variation due to random choice of dataset and, for finite nets, random initialization. **(A)** Learnabilities of sinusoidal eigenfunctions on the unit circle discretized into $M = 2^8$ points. Eigenfunctions with higher $k$ have lower eigenvalues and thus require more data to learn. At $n = 2^8$, the training set contains all input points and all functions are thus predicted perfectly. **(B)** Learnabilities of subset-parity functions on the vertices of the 8d hypercube. Eigenfunctions with higher $k$ again have lower eigenvalues and are learned later, with all functions predicted perfectly at $n = 2^8$. The dashed line indicates $\mathcal{L} = n/M$, the learnability of any function with respect to a random model; $\mathcal{L}(\phi_8)$ falls below this curve, showing that the $k = 8$ eigenmode generalizes worse than chance. **(C)** Learnabilities of hyperspherical harmonics on the continuous 7-sphere $\mathbb{S}^7$. Eigenfunctions with higher $k$ again have lower eigenvalues and are learned later, but the continuous input space prevents learnability from exactly reaching 1.

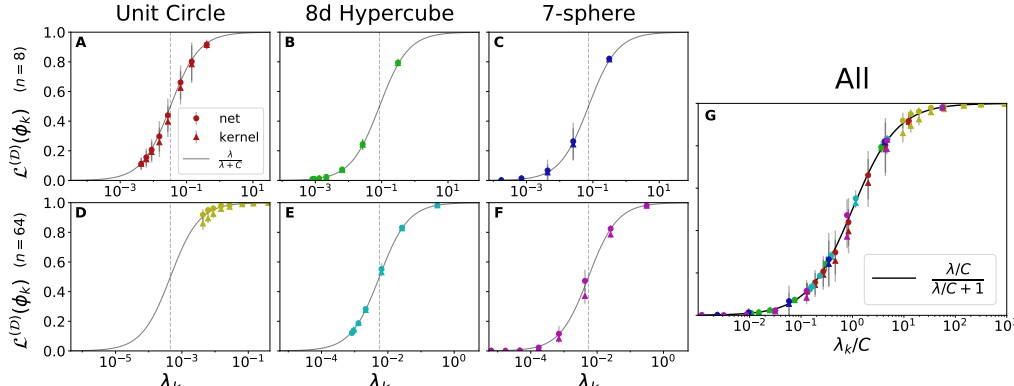

Figure 2: **Eigenmode learnability vs. eigenvalue takes a universal functional form.** For any dataset size and input domain, eigenmode learnability closely follows a universal curve $\lambda/(\lambda + C)$ with one problem-dependent parameter $C$. Theoretical curves (solid lines) have the same sigmoidal shape in every panel. True eigenmode learnabilities $\mathcal{L}(\phi_k)$ for $k \in \{0, .., 7\}$ for both exact NTK regression (triangles) and finite networks (circles) exhibit excellent agreement. Vertical dashed lines indicate $C$ for each learning problem. **(A-C)** Learnability vs. eigenvalue for eigenmodes of the unit circle, 8d hypercube, and 7-sphere with $n = 8$. The $k \in \{4, ..., 7\}$ modes are cut off to the left of (C). **(D-F)** Learnability curves with $n = 64$. Eigenmodes lie higher on each curve than the corresponding points in (A-C), reflecting greater learnability due to the larger $n$. **(G)** All points from (A-F), rescaled by their respective values of $C$, lie on the same universal curve $\frac{\lambda/C}{\lambda/C+1}$.

**Predicting learnability $\mathcal{L}^{(\mathcal{D})}$.** We use both wide finite nets and exact NTK regression to learn several eigenmodes on all three domains and compare true learnabilities with our theoretical predictions. Our theory predicts true generalization behavior quite well (Figure 1). We further show that the $k = 8$ mode on the $8d$ hypercube has worse-than-chance generalization, an inevitable result of its low eigenvalue and Corollary 1. We note that finite networks and NTK regression give very similar results, supporting the validity of approximating networks with the NTK.

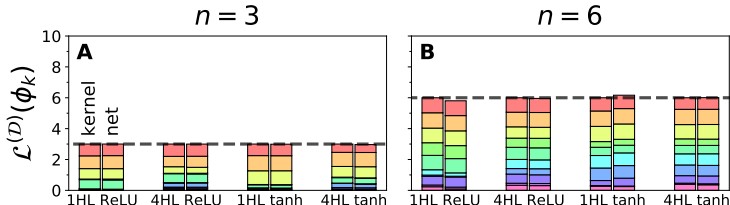

Figure 3: **Eigenfunction learnability always sums to the size of the training set.** Stacked bar charts show $\mathcal{D}$-learnability for a particular random $\mathcal{D}$ for each of the 10 eigenfunctions over a simple 10-point domain. For all architectures, the total height of each bar is approximately $n$. **(A)** As per Theorem 1, the summed $\mathcal{D}$-learnabilities for exact NTK regression (left bars) are all exactly $n$, and those for trained finite nets (right bars) are remarkably close. Stacked bars show $\mathcal{D}$-learnability for the 10 eigenfunctions, all from the same training set $\mathcal{D}$ of $n = 3$ data points, stacked from top to bottom in descending order of eigenvalue. A different network architecture was used in each of the four pairs of columns. As per Lemma 1d, the height of each eigenmode contribution falls in $[0, 1]$. **(B)** Same as (A) with $n = 6$.

**Universal form for learnability.** We next fix $n$ and plot learnability vs. eigenvalue for eight modes over each domain. In each case we find $\mathcal{L}(\phi) \approx \lambda/(\lambda + C)$, and by rescaling the eigenvalues (a symmetry which leaves Equation 1 invariant), we see that for any neural network learning problem with any training set size $n$, eigenmode learnability always lies on one universal curve (Figure 2).

**No free lunch for neural networks.** We then experimentally confirm that our no-free-lunch result applies to finite networks as well as kernel regression (Figure 3). We use both models to learn a function on the discretized unit circle with $M = 10$ and sum the resulting $\mathcal{D}$-learnabilities of each eigenmode. Unlike in our other experiments, we only sample $\mathcal{D}$ once for each $n$; even without the benefit of averaging over datasets, we find that total $\mathcal{D}$-learnability is always conserved.

**Nonmonotonic MSE curves.** We next confirm our first-principles prediction of nonmonotonic MSE curves at small $n$. We plot curves for four eigenmodes on each domain, three of which Equation 16 predicts will have positive $d\mathcal{E}/dn|_{n=0}$. MSE is indeed increasing as predicted (Figure A1).

**Agreement with narrow networks.** Finally, we repeat the experiment of Figure 1B for varying network width. We find that our theory gives accurate predictions of learnability and MSE even for networks of depth 4 and width as small as 20 (Figures B2 and B3). This surprising agreement suggests our theory is a promising starting point for understanding the generalization of practical neural networks.

## 4 CONCLUSION

We have presented a first-principles theory of generalization for kernel regression that predicts a variety of measures of generalization performance. We then empirically demonstrated its predictive accuracy for both exact NTK regression and wide finite networks in the NTK regime. This theory offers new insight into these models' inductive bias and provides a general framework for understanding their learning behavior, opening the door to the principled study of many other deep learning mysteries.

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

## A  Nonmonotonic MSE curves at small training set size

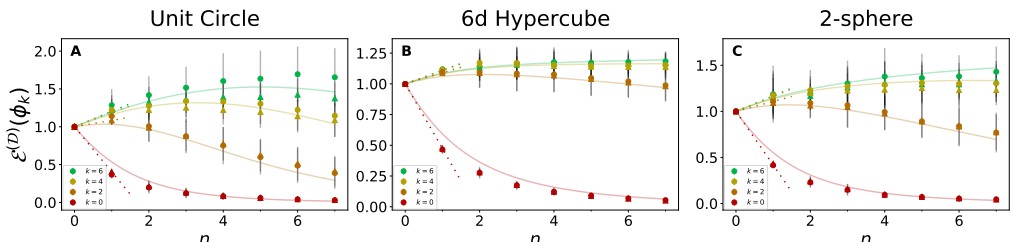

Figure A1: **Our theory correctly predicts that, for low-eigenvalue eigenfunctions, MSE counterintuively *increases* as points are added to a small training set. (A-C)** Generalization MSE of exact NTK regression (triangles) and finite networks (circles) when learning four different eigenmodes on each of three different domains given $n$ training points. Theoretical curves closely match experimental data. Eigenmodes with higher $k$ have lower eigenvalues and thus higher mean MSEs, and for $k \in \{2, 4, 6\}$, MSE even *increases* as $n$ increases from zero. Dashed lines show $d\mathcal{E}/dn|_{n=0}$ as predicted by Equation 16.

## B  Experimental results for narrow networks

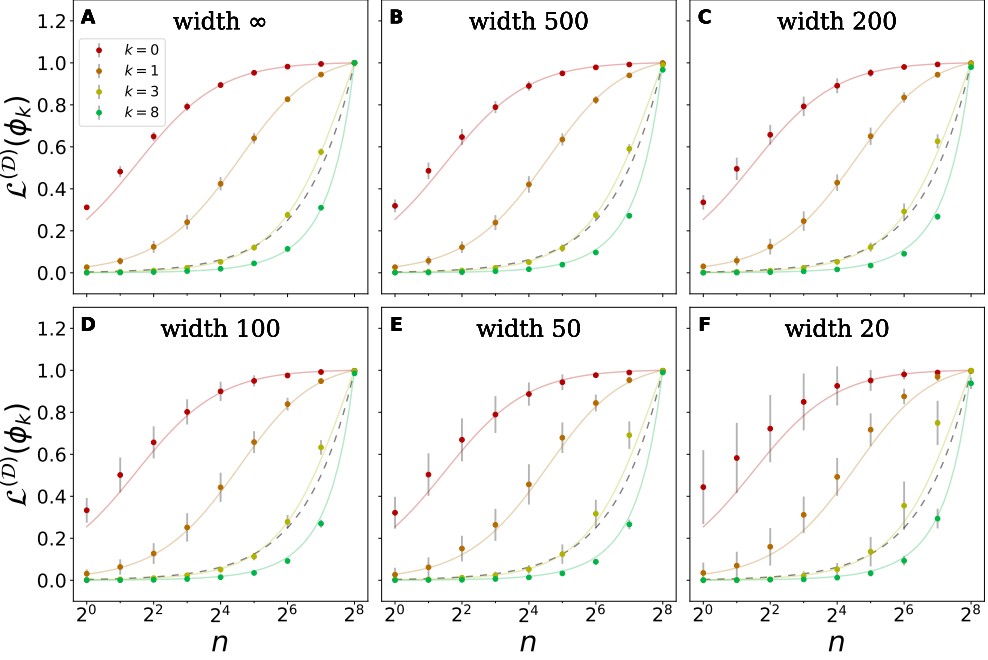

Figure B2: **Theoretical learnability predictions remain accurate even for quite narrow networks of moderate depth.** Plots show learnability vs. training set size for four eigenmodes of the 8d hypercube, learned with a 4L ReLU net with various widths. Except for changing width, these experiments are identical to those of Figure 1b. Theoretical predictions (solid curves) are the same in all plots. Dashed lines show learnability from a naive, nongeneralizing model; points below the line imply worse-than-chance generalization (see Corollary 1). **(A)** Infinite-width results using exact NTK regression. **(B-F)** Results for successively narrower finite networks. As width decreases, mean learnability increases slightly, and $1\sigma$ error grows. Despite this, mean learnabilities remain remarkably close to our theoretical predictions even at width 20.

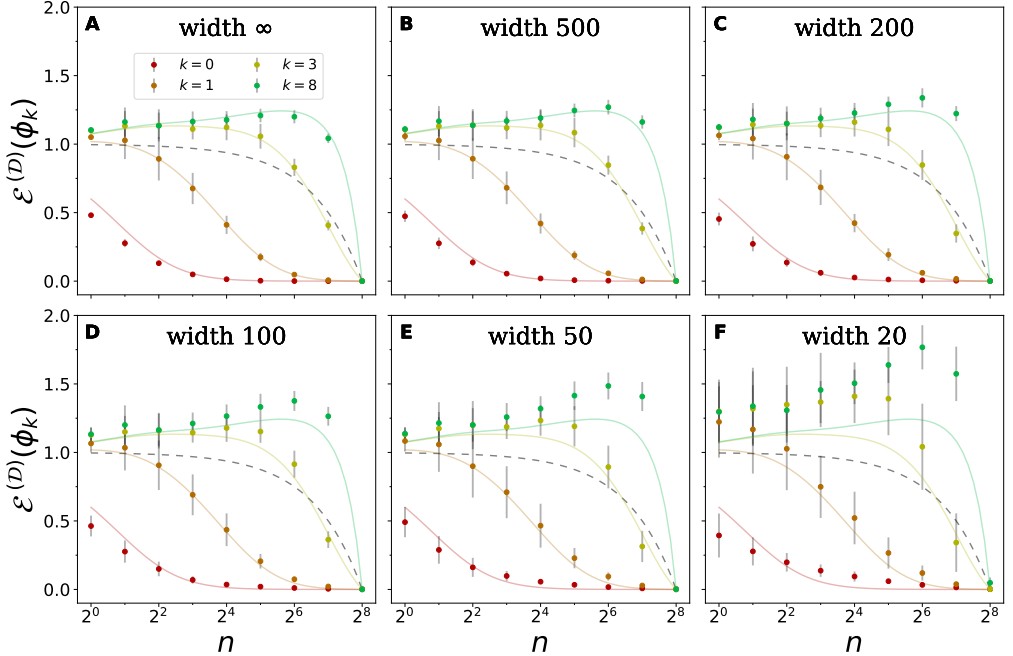

Figure B3: **Theoretical MSE predictions remain accurate even for quite narrow networks.** Plots show MSE vs. training set size for four eigenmodes of the 8d hypercube, learned with a 4L ReLU net with various widths. Except for changing width and the fact that MSE is plotted instead of learnability, these experiments are identical to those of Figure 1b. Theoretical predictions (solid curves) are the same in all plots. Dashed lines show MSE from a naive, nongeneralizing model; points above the dashed lines imply worse-than-chance generalization (see Theorem [THEOREM]). **(A)** Infinite-width results using exact NTK regression. **(B-F)** Results for successively narrower finite networks. As width decreases, MSE tends to increase, but only slightly. Theoretical predictions remain remarkably accurate for most eigenmodes down to width 50, with qualitative agreement even at width 20.

## C  REVIEW OF THE NTK

In the main text, we assume prior familiarity with the NTK, using Equation 1 as the starting point of our derivations. Here we provide a definition and very brief introduction to the NTK for unfamiliar readers. For derivations and full discussions, see Jacot et al. (2018) and Lee et al. (2019).

Consider a feedforward neural network representing a function $\hat{f}_\theta : \mathcal{X} \to \mathbb{R}$, where $\theta$ is a parameter vector. Further consider one training example $x$ with target value $y$ and one test point $x'$ and suppose we perform one step of gradient descent with a small learning rate $\eta$ with respect to the MSE loss $\ell_\theta \equiv (\hat{f}_\theta(x) - y)^2$. This gives the parameter update

$$\theta \to \theta + \delta\theta, \quad \text{with} \quad \delta\theta = -\eta\nabla_\theta\ell_\theta = -2\eta(\hat{f} - y)\nabla_\theta\hat{f}_\theta(x). \tag{17}$$

We now wish to know how this parameter update changes $\hat{f}_\theta(x')$. To do so, we linearize about $\theta$, finding that

$$\begin{aligned}
\hat{f}_{\theta+\delta\theta}(x') &= \hat{f}_\theta(x') + \nabla_\theta\hat{f}_\theta(x') \cdot \delta\theta + \mathcal{O}(\delta\theta^2) \\
&= \hat{f}_\theta(x') - 2\eta(\hat{f} - y)\Big[\nabla_\theta\hat{f}_\theta(x) \cdot \nabla_\theta\hat{f}_\theta(x')\Big] + \mathcal{O}(\delta\theta^2) \\
&= \hat{f}_\theta(x') - 2\eta(\hat{f} - y)K(x, x') + \mathcal{O}(\delta\theta^2),
\end{aligned} \tag{18}$$

where we have defined $K(x, x') \equiv \nabla_\theta \hat{f}_\theta(x) \cdot \nabla_\theta \hat{f}_\theta(x')$. This quantity is the NTK. Remarkably, as network width[6] goes to infinity, the $\mathcal{O}(\delta\theta^2)$ corrections become negligible, and $K(x, x')$ is the same after any random initialization[7] and at any time during training. This dramatically simplifies the analysis of network training, allowing one to prove that after infinite time training on MSE loss for an arbitrary dataset, the network's learned function is given by Equation 1. See, for example, Equations 14-16 of Lee et al. (2019)[8].

# D  PROOF OF LEMMA 1

**Property (a)**: $\mathcal{L}^{(\mathcal{D})}(\phi_i) = \mathbf{T}_{ii}^{(\mathcal{D})}$, and $\mathcal{L}(\phi_i) = \mathbf{T}_{ii}$.

*Proof.* Using the fact that $\langle \phi_i, \phi_i \rangle = 1$, we see that $\mathcal{L}^{(\mathcal{D})}(\phi_i) = \langle \hat{\phi}_i, \phi_i \rangle = \mathbf{e}_i^T \mathbf{T}^{(\mathcal{D})} \mathbf{e}_i = \mathbf{T}_{ii}^{(\mathcal{D})}$, where $\mathbf{e}_i$ is a one-hot $M$-vector with the one at index $i$. The second clause of the property follows by averaging.

**Property (b)**: When $n = 0$, $\mathbf{T} = \mathbf{T}^{(\mathcal{D})} = 0$ and $\mathcal{L}(f) = \mathcal{L}^{(\mathcal{D})}(f) = 0$.

*Proof.* When $n = 0$, $\mathbf{\Phi}(\mathcal{D})$ has no columns, and thus $\mathbf{T}^{(\mathcal{D})} = 0$. The other clauses follow from Property (a) and averaging.

**Property (c)**: When $n = M$, $\mathbf{T} = \mathbf{T}^{(\mathcal{D})} = \mathbf{I}_M$ and $\mathcal{L}(f) = \mathcal{L}^{(\mathcal{D})}(f) = 1$.

*Proof.* When $n = M$, $\mathbf{\Phi}(\mathcal{D})$ is a full-rank $M \times M$ matrix. Inspection of the formula for $\mathbf{T}^{(\mathcal{D})}$ then shows that $\mathbf{T}^{(\mathcal{D})} = \mathbf{I}_M$. The other clauses follow from Property (a) and averaging.

**Property (d)**: All eigenvalues of $\mathbf{T}^{(\mathcal{D})}$ are in $\{0, 1\}$, all eigenvalues of $\mathbf{T}$ are in $[0, 1]$, and so $\mathcal{L}(f), \mathcal{L}^{(\mathcal{D})}(f) \in [0, 1]$.

*Proof.* From the definition of $\mathbf{T}^{(\mathcal{D})}$, it is easy to see that $\mathbf{T}^{(\mathcal{D})} = \left( \mathbf{T}^{(\mathcal{D})} \right)^2$. $\mathbf{T}^{(\mathcal{D})}$ is thus idempotent, with all eigenvalues in $\{0, 1\}$. The fact that all eigenvalues of $\mathbf{T}$ are in $[0, 1]$ follows by averaging. The stated properties of $\mathcal{L}(f), \mathcal{L}^{(\mathcal{D})}(f)$ follow from the fact that, for any compatible vector $\mathbf{z}$ and matrix $\mathbf{A}$, it holds that $\frac{\mathbf{z}^T A \mathbf{z}}{\mathbf{z}^T \mathbf{z}}$ is bounded by the maximum and minimum eigenvalues of $\mathbf{A}$.

**Property (e)**: Let $\mathcal{D}_+$ be $\mathcal{D} \cup x$, where $x \in X, x \notin \mathcal{D}$ is a new data point. Then $\mathcal{L}^{(\mathcal{D}_+)}(f) \geq \mathcal{L}^{(\mathcal{D})}(f)$.

*Proof.* To begin, we use the Moore-Penrose pseudoinverse, which we denote by $(\cdot)^+$, to cast $\mathbf{T}^{(\mathcal{D})}$ into a more transparent form:

$$\mathbf{T}^{(\mathcal{D})} \equiv \mathbf{\Lambda}\mathbf{\Phi} \left( \mathbf{\Phi}^T \mathbf{\Lambda} \mathbf{\Phi} \right)^{-1} \mathbf{\Phi}^T = \mathbf{\Lambda}^{1/2} \left( \mathbf{\Lambda}^{1/2} \mathbf{\Phi}\mathbf{\Phi}^T \mathbf{\Lambda}^{1/2} \right) \left( \mathbf{\Lambda}^{1/2} \mathbf{\Phi}\mathbf{\Phi}^T \mathbf{\Lambda}^{1/2} \right)^+ \mathbf{\Lambda}^{-1/2}, \quad (19)$$

where we have suppressed the $\mathcal{D}$ in $\mathbf{\Phi}(\mathcal{D})$. This follows from the property of pseudoinverses that $\mathbf{A}(\mathbf{A}^T \mathbf{A})^+ \mathbf{A}^T = (\mathbf{A}\mathbf{A}^T)(\mathbf{A}\mathbf{A}^T)^+$ for any matrix $\mathbf{A}$. We now augment our system with one extra data point, getting

$$\mathbf{T}^{(\mathcal{D}_+)} = \mathbf{\Lambda}^{1/2} \left( \mathbf{\Lambda}^{1/2}(\mathbf{\Phi}\mathbf{\Phi}^T + \xi\xi^T)\mathbf{\Lambda}^{1/2} \right) \left( \mathbf{\Lambda}^{1/2}(\mathbf{\Phi}\mathbf{\Phi}^T + \xi\xi^T)\mathbf{\Lambda}^{1/2} \right)^+ \mathbf{\Lambda}^{-1/2}, \quad (20)$$

where $\xi$ is an $M$-element column vector orthogonal to the others of $\mathbf{\Phi}$. We now convert the pseudoinverse into an inverse with a limit, getting

$$\mathbf{T}^{(\mathcal{D}_+)} = \lim_{\delta \to 0^+} \mathbf{\Lambda}^{1/2} \left( \mathbf{\Lambda}^{1/2}(\mathbf{\Phi}\mathbf{\Phi}^T + \xi\xi^T)\mathbf{\Lambda}^{1/2} \right) \left( \mathbf{\Lambda}^{1/2}(\mathbf{\Phi}\mathbf{\Phi}^T + \xi\xi^T)\mathbf{\Lambda}^{1/2} + \delta\mathbf{I}_M \right)^{-1} \mathbf{\Lambda}^{-1/2}. \quad (21)$$

---

[6]The "width" parameter varies by architecture; for example it is the minimal hidden layer width for fully connected networks and the minimal number of channels per hidden layer for a convolutional network.

[7](assuming the parameters are drawn from the same distribution)

[8]We note that there exists a different infinite-width kernel, called the "NNGP kernel," describing a network's random initialization, and this reference uses $\mathcal{K}$ for the NNGP kernel and $\Theta$ for the NTK.

We now use the Sherman-Morrison matrix inversion formula to find that

$$\mathbf{T}^{(\mathcal{D}_+)} = \mathbf{T}^{(\mathcal{D})} + \lim_{\delta \to 0^+} \delta \frac{\left(\mathbf{\Lambda}^{1/2}\mathbf{\Phi}\mathbf{\Phi}^T\mathbf{\Lambda}^{1/2} + \delta\mathbf{I}_M\right)^{-1} \xi\xi^T \left(\mathbf{\Lambda}^{1/2}\mathbf{\Phi}\mathbf{\Phi}^T\mathbf{\Lambda}^{1/2} + \delta\mathbf{I}_M\right)^{-1}}{1 + \xi^T \left(\mathbf{\Lambda}^{1/2}\mathbf{\Phi}\mathbf{\Phi}^T\mathbf{\Lambda}^{1/2} + \delta\mathbf{I}_M\right)^{-1}\xi}. \tag{22}$$

Because both $\xi\xi^T$ and the inverted matrix in Equation 22 are positive semidefinite, we conclude that, for any $M$-vector $\mathbf{z}$, it will hold that $\mathbf{z}^T\mathbf{T}^{(\mathcal{D}_+)}\mathbf{z} \geq \mathbf{z}^T\mathbf{T}^{(\mathcal{D})}\mathbf{z}$. The desired property follows.

***Property (f)***: For any $i \in \{1, ..., M\}$, $\frac{d}{d\lambda_i}\mathbf{T}_{ii}^{(\mathcal{D})} \geq 0$, hence $\frac{d}{d\lambda_i}\mathcal{L}^{(\mathcal{D})}(\phi_i) \geq 0$.

*Proof.* Differentiating $\mathbf{T}_{jj}^{(\mathcal{D})}$ with respect to a particular $\lambda_i$, we find that

$$\frac{d}{d\lambda_i}\mathbf{T}_{jj}^{(\mathcal{D})} = (\delta_{ij} - \lambda_j\phi_j^T K^{-1}\phi_i)\phi_i^T K^{-1}\phi_j, \tag{23}$$

where $\phi_i^T$ is the $i$-th row of $\mathbf{\Phi}$ and $K = \mathbf{\Phi}^T\mathbf{\Lambda}\mathbf{\Phi}$. Specializing to the case $i = j$, we note that $\phi_i^T K^{-1}\phi_i \geq 0$ because $K$ is positive semidefinite, and $\lambda_j\phi_i K^{-1}\phi_i^T \leq 1$ because $\lambda_j\phi_i\phi_i^T$ is one of the positive semidefinite summands in $K = \sum_k \lambda_k\phi_k\phi_k^T$. The desired property follows.

***Property (g)***: For any $i, j \in \{1, ..., M\}$, $i \neq j$, $\frac{d}{d\lambda_i}\mathbf{T}_{jj}^{(\mathcal{D})} \leq 0$, hence $\frac{d}{d\lambda_i}\mathcal{L}^{(\mathcal{D})}(\phi_j) \leq 0$.

*Proof.* Differentiating as in the proof of Property (f) and using the fact that $i \neq j$, we see that

$$\frac{d}{d\lambda_i}\mathbf{T}_{jj}^{(\mathcal{D})} = -\lambda_j \left(\phi_j^T K^{-1}\phi_i\right)^2, \tag{24}$$

which is manifestly nonpositive because $\lambda_j > 0$. The desired property follows.

## E  PROOFS OF THEOREM 1 AND COROLLARY 1

Here we prove Theorem 1, our "no-free-lunch" result, and Corollary 1, which states the existence of eigenfunctions that generalize worse than chance. We begin with the former.

*Proof of Theorem 1.* First, we note that, for any orthogonal basis $\mathcal{F}$ on $\mathcal{X}$,

$$\sum_{f \in \mathcal{F}} \mathcal{L}^{(\mathcal{D})}(f) = \sum_{\mathbf{v} \in \mathcal{V}} \frac{\mathbf{v}^T\mathbf{T}^{(\mathcal{D})}\mathbf{v}}{\mathbf{v}^T\mathbf{v}}, \tag{25}$$

where $\mathcal{V}$ is an orthogonal set of vectors spanning $\mathbb{R}^M$. This is equivalent to $\text{Tr}\left[\mathbf{T}^{(\mathcal{D})}\right]$. This trace is given by

$$\text{Tr}\left[\mathbf{T}^{(\mathcal{D})}\right] = \text{Tr}\left[\mathbf{\Phi}^T\mathbf{\Lambda}\mathbf{\Phi}(\mathbf{\Phi}^T\mathbf{\Lambda}\mathbf{\Phi})^{-1}\right] = \text{Tr}[\mathbf{I}_n] = n, \tag{26}$$

which proves the desired theorem. $\square$

*Proof of Corollary 1.* First, we note that the naive, nongeneralizing model described in the theorem will always have a learnability of $\mathcal{L}_{naive}(f) = \mathbb{E}_{\mathcal{D}}\left[\frac{n\langle f, f\rangle}{M\langle f, f\rangle}\right] = \frac{n}{M}$. As per Theorem 1, this is exactly the *mean* learnability of all eigenfunctions, so either all have precisely this learnability or else one has lower learnability. This proves the clause of the theorem specific to learnability.

To prove the clause specific to MSE, we now note that, as defined by Equation 1, kernel regression will always perfectly memorize the training data, so for both the naive model and kernel regression, MSE is given by

$$\mathcal{E}(f) = \frac{1}{M}\sum_{x \in \mathcal{X} \setminus \mathcal{D}}(f(x) - \hat{f}(x))^2 = \frac{1}{M}\sum_{x \in \mathcal{X} \setminus \mathcal{D}}f^2(x) + \frac{1}{M}\sum_{x \in \mathcal{X} \setminus \mathcal{D}}\hat{f}(x)^2 - \frac{2}{M}\sum_{x \in \mathcal{X} \setminus \mathcal{D}}f(x)\hat{f}(x). \quad (27)$$

The first term is the same for any model, the second term is nonnegative but zero for the naive model, and the third term is equivalent to $-2(\mathcal{L}(f) - \frac{n}{M})$, which must be nonnegative for at least one eigenfunction. Kernel regression therefore gives worse MSE than the naive model unless, as for the naive model, $\hat{f}(x) = 0$ for $x \notin \mathcal{D}$. $\square$

## F  Approximating $\mathbf{T}$

Here we provide details of the derivations of our approximation for $\mathbf{T}$ and of Lemma 2. Whenever possible, we leave it implicit that all expressions for $\mathbf{T}$ in this appendix will be approximations and use the symbol for equality for simplicity.

We begin by taking the approximation of Equation 8 that

$$\mathbf{T} \approx \mathbb{E}_{\mathbf{\Phi}}\left[\mathbf{\Lambda}\mathbf{\Phi}\left(\mathbf{\Phi}^T\mathbf{\Lambda}\mathbf{\Phi}\right)^{-1}\mathbf{\Phi}^T\right], \quad (28)$$

where the expectation is taken over all $M \times n$ matrices $\mathbf{\Phi}$ satisfying $\mathbf{\Phi}^T\mathbf{\Phi} = M\mathbf{1}_n$ with the uniform measure. It turns out that we can equivalently average over all *all* $M \times n$ matrices $\mathbf{\Phi}$ with a mean-zero i.i.d. Gaussian measure over each element, without changing $\mathbf{T}$. To see this, note that this equation is symmetric under right-multiplication of $\mathbf{\Phi}$ by arbitrary $n \times n$ invertible matrices $\mathbf{R}$:

$$\mathbf{\Lambda}\mathbf{\Phi}\left(\mathbf{\Phi}^T\mathbf{\Lambda}\mathbf{\Phi}\right)^{-1}\mathbf{\Phi}^T = \mathbf{\Lambda}\mathbf{\Phi}\mathbf{R}\left(\mathbf{R}^T\mathbf{\Phi}^T\mathbf{\Lambda}\mathbf{\Phi}\mathbf{R}\right)^{-1}\mathbf{R}^T\mathbf{\Phi}^T. \quad (29)$$

Letting $\mathbf{R}$ always be the ($\mathbf{\Phi}$-dependent) matrix that orthogonalizes the columns of $\mathbf{\Phi}$ (i.e. $\mathbf{\Phi}^T\mathbf{R}^T\mathbf{R}\mathbf{\Phi} = M\mathbf{I}_n$), one can see that, no matter the distribution of $\mathbf{\Phi}$ in Equation 28, it holds that $\mathbb{E}_{\mathbf{\Phi}}\left[\mathbf{\Lambda}\mathbf{\Phi}\left(\mathbf{\Phi}^T\mathbf{\Lambda}\mathbf{\Phi}\right)^{-1}\mathbf{\Phi}^T\right] = \mathbb{E}_{\mathbf{\Phi}}\left[\mathbf{\Lambda}\mathbf{\Phi}_o\left(\mathbf{\Phi}_o^T\mathbf{\Lambda}\mathbf{\Phi}_o\right)^{-1}\mathbf{\Phi}_o^T\right]$, where $\mathbf{\Phi}_o$ is the orthogonalized version of $\mathbf{\Phi}$. The expectation can thus equivalently be taken over any distribution that induces the same distribution over the orthogonalized $\mathbf{\Phi}_o$; in particular, the uniform measure on orthogonal $\mathbf{\Phi}$ can be interchanged with the standard Gaussian measure. Moving forward, we will exploit this equivalence and assume that each element of $\mathbf{\Phi}$ is sampled i.i.d. from $\mathcal{N}(0,1)$. This assumption will largely remain in the background, but will be useful later.

We now evaluate $\mathbf{T}$, starting with the off-diagonal elements. We observe that

$$\mathbb{E}_{\mathbf{\Phi}}\left[\mathbf{\Lambda}\mathbf{\Phi}\left(\mathbf{\Phi}^T\mathbf{U}^T\mathbf{\Lambda}\mathbf{U}\mathbf{\Phi}\right)^{-1}\mathbf{\Phi}^T\right] = \mathbb{E}_{\mathbf{\Phi}}\left[\mathbf{\Lambda}\mathbf{U}^T\mathbf{\Phi}\left(\mathbf{\Phi}^T\mathbf{\Lambda}\mathbf{\Phi}\right)^{-1}\mathbf{\Phi}^T\mathbf{U}\right], \quad (30)$$

where $\mathbf{U}$ is any orthogonal $M \times M$ matrix. Defining $\mathbf{U}^{(m)}$ as the matrix such that $\mathbf{U}^{(m)}_{ab} \equiv \delta_{ab}(1 - 2\delta_{am})$, noting that $\mathbf{U}^{(m)}\mathbf{\Lambda}\mathbf{U}^{(m)} = \mathbf{\Lambda}$, and plugging $\mathbf{U}^{(m)}$ in as $\mathbf{U}$ in Equation 30, we find that

$$\mathbf{T}_{ab} = \left(\left(\mathbf{U}^{(m)}\right)^T\mathbf{T}\mathbf{U}^{(m)}\right)_{ab} = (-1)^{\delta_{am}+\delta_{bm}}\mathbf{T}_{ab}. \quad (31)$$

By choosing $m = a$, we conclude that $\mathbf{T}_{ab} = 0$ if $a \neq b$.

To evaluate the diagonal elements of $\mathbf{T}$, we consider augmenting our eigensystem with a new "test eigenmode" with eigenvalue $\lambda$, as described in the text and formalized in Equation 9. To proceed, we make the core assumption that $\mathbf{T}_{ii} \approx \mathbf{T}_{ii}^+$ when $n, M \gg 1$ [9]. This assumption, combined with the symmetry $\mathbf{T}_{ii}^+ = \mathbf{T}_{jj}^+$ whenever $\lambda_i = \lambda_j$, implies that $\mathbf{T}_{ii} = \mathbf{T}_{(M+1)(M+1)}^+|_{\lambda=\lambda_i}$. Therefore, to evaluate $\mathbf{T}$, it suffices to evaluate $\mathbf{T}_{(M+1)(M+1)}^+$ as a function of $\lambda$.

---

[9]Empirically, we see that this approximation holds already for realistic values of $n$ and $M$ used in neural net experiments.

We now manipulate the expression for $\mathbf{T}^+$ to isolate $\mathbf{T}^+_{(M+1)(M+1)}$. Using the Sherman-Morrison matrix inversion formula, we obtain

$$
\begin{aligned}
\left(\mathbf{\Phi}^{+T}\mathbf{\Lambda}^+\mathbf{\Phi}^+\right)^{-1} &= \left(\mathbf{\Phi}^T\mathbf{\Lambda}\mathbf{\Phi} + \lambda\phi\phi^T\right)^{-1} \\
&= \left(\mathbf{\Phi}^T\mathbf{\Lambda}\mathbf{\Phi}\right)^{-1} - \frac{\lambda\left(\mathbf{\Phi}^T\mathbf{\Lambda}\mathbf{\Phi}\right)^{-1}\phi\phi^T\left(\mathbf{\Phi}^T\mathbf{\Lambda}\mathbf{\Phi}\right)^{-1}}{1 + \lambda\phi^T\left(\mathbf{\Phi}^T\mathbf{\Lambda}\mathbf{\Phi}\right)^{-1}\phi},
\end{aligned}
\tag{32}
$$

and inserting this into the full expression for $\mathbf{T}^+$ gives

$$
\begin{aligned}
\mathbf{T}^+_{(M+1)(M+1)} &= \mathbb{E}_{\mathbf{\Phi},\phi}\left[\lambda\phi^T\left(\mathbf{\Phi}^{+T}\mathbf{\Lambda}\mathbf{\Phi}^+ + \lambda\phi\phi^T\right)^{-1}\phi\right] \\
&= \mathbb{E}_{\mathbf{\Phi},\phi}\left[\lambda\phi^T\left(\mathbf{\Phi}^T\mathbf{\Lambda}\mathbf{\Phi}\right)^{-1}\phi - \frac{\lambda^2\left[\phi^T\left(\mathbf{\Phi}^T\mathbf{\Lambda}\mathbf{\Phi}\right)^{-1}\phi\right]^2}{1 + \lambda\phi^T\left(\mathbf{\Phi}^T\mathbf{\Lambda}\mathbf{\Phi}\right)^{-1}\phi}\right] \\
&= \mathbb{E}_{\mathbf{\Phi},\phi}\left[\frac{\lambda}{\lambda + \left[\phi^T\left(\mathbf{\Phi}^T\mathbf{\Lambda}\mathbf{\Phi}\right)^{-1}\phi\right]^{-1}}\right] \\
&= \mathbb{E}_{\mathbf{\Phi},\phi}\left[\frac{\lambda}{\lambda + C^{(\mathbf{\Phi},\phi)}}\right],
\end{aligned}
\tag{33}
$$

where $C^{(\mathbf{\Phi},\phi)} \equiv \left[\phi^T\left(\mathbf{\Phi}^T\mathbf{\Lambda}\mathbf{\Phi}\right)^{-1}\phi\right]^{-1}$ is a nonnegative scalar.

We now argue that, for realistic values of $M$, $n$, and $\{\lambda_i\}_i$, we can simply replace $C^{(\mathbf{\Phi},\phi)}$ by its mean. We first note that, because $\lambda/(\lambda + C)$ is convex, we can use Jensen's inequality to find that

$$
\mathbb{E}_{\mathbf{\Phi},\phi}\left[\frac{\lambda}{\lambda + C^{(\mathbf{\Phi},\phi)}}\right] \geq \frac{\lambda}{\lambda + \bar{C}},
\tag{34}
$$

where we have defined $\bar{C} \equiv \mathbb{E}_{\mathbf{\Phi},\phi}\left[C^{(\mathbf{\Phi},\phi)}\right]$. Some algebra also yields the fact that

$$
\frac{\lambda}{\lambda + C^{(\mathbf{\Phi},\phi)}} \leq \frac{\lambda}{\lambda + \bar{C}} - \frac{\lambda(C^{(\mathbf{\Phi},\phi)} - \bar{C})}{(\lambda + \bar{C})^2} + \frac{(C^{(\mathbf{\Phi},\phi)} - \bar{C})^2}{(\lambda + \bar{C})^2},
\tag{35}
$$

for $C^{(\mathbf{\Phi},\phi)} \geq 0$, which, after taking an expectation, gives us the upper bound that

$$
\begin{aligned}
\mathbb{E}_{\mathbf{\Phi},\phi}\left[\frac{\lambda}{\lambda + C^{(\mathbf{\Phi},\phi)}}\right] &\leq \frac{\lambda}{\lambda + \bar{C}} + \frac{\mathrm{Var}_{\mathbf{\Phi},\phi}\left[C^{(\mathbf{\Phi},\phi)}\right]}{(\lambda + \bar{C})^2} \\
&\leq \frac{\lambda}{\lambda + \bar{C}} + \frac{\mathrm{Var}_{\mathbf{\Phi},\phi}\left[C^{(\mathbf{\Phi},\phi)}\right]}{\bar{C}^2}.
\end{aligned}
\tag{36}
$$

Armed with the bounds of Equation 34 and Equation 36, we can replace $C^{(\mathbf{\Phi},\phi)}$ by $\bar{C}$ in the last line of Equation 33 if we can show that $\mathrm{Var}_{\mathbf{\Phi},\phi}\left[C^{(\mathbf{\Phi},\phi)}\right] \ll \bar{C}^2$. This will be nontrivial to show in general, but we can easily see that it holds in a caricature of a realistic problem. In practice, we often find that the NTK spectrum is dominated by a few large eigenvalues with a long tail of small eigenvalues (Figure I5). To model this sort of spectrum, let $p \in \mathbb{N}$ with $p \ll n \ll M$, and suppose

$$
\lambda_i = \begin{cases} \lambda_0 & \text{if } 1 \leq i \leq p \\ \frac{\lambda'}{M-p} & \text{if } p+1 \leq i \leq M \end{cases},
\tag{37}
$$

with $\lambda_0 \gg \lambda'/(M-p)$. The kernel matrix is given by

$$\mathbf{\Phi}^T \mathbf{\Lambda} \mathbf{\Phi} = \sum_{i=1}^{p} \lambda_0 \phi_i \phi_i^T + \sum_{i=p+1}^{M} \frac{\lambda'}{M-p} \phi_i \phi_i^T \approx \sum_{i=1}^{p} \lambda_0 \phi_i \phi_i^T + \lambda' \mathbf{I}_n, \tag{38}$$

where $\phi_i$ is the $i$-th row of $\mathbf{\Phi}$ and we have made an approximation using the central limit theorem assuming that $\mathbf{\Phi}$ is random with entries sampled i.i.d. from $\mathcal{N}(0,1)$. This kernel matrix has $p$ large, $\mathcal{O}(\lambda_0)$ eigenvalues and $n-p$ remaining eigenvalues of order $\lambda'$. Its inverse therefore has $p$ small eigenvalues and $n-p$ similar large ones of order $1/\lambda'$. Letting $\{\xi_i\}_{i=1}^{n}$ be these inverted eigenvalues, we have that

$$C^{(\mathbf{\Phi},\phi)} = \frac{1}{\phi^T \left(\mathbf{\Phi}^T \mathbf{\Lambda} \mathbf{\Phi}\right)^{-1} \phi} = \frac{1}{\sum_{i=1}^{n} \xi_i x_i^2}, \tag{39}$$

where $\{x_i\}_{i=1}^{n}$ are the components of $\phi$ on each of the kernel eigendirections. For typical $\{x_i\}_{i=1}^{n}$ sampled i.i.d. from $\mathcal{N}(0,1)$, the denominator approaches $(n-p)\lambda'$ by the central limit theorem, and thus we reach our desired conclusion that $C^{(\mathbf{\Phi},\phi)} \approx 1/((n-p)\lambda')$ is approximately deterministic. More precisely, we find

$$\mathbb{E}_{\mathbf{\Phi},\phi}\left[C^{(\mathbf{\Phi},\phi)}\right] \approx \frac{1}{(n-p)\lambda'}, \quad \mathrm{Var}_{\mathbf{\Phi},\phi}\left[C^{(\mathbf{\Phi},\phi)}\right] \approx \frac{3}{(n-p)^3 (\lambda')^2}, \tag{40}$$

from which we see that

$$\frac{\mathrm{Var}_{\mathbf{\Phi},\phi}\left[C^{(\mathbf{\Phi},\phi)}\right]}{\bar{C}^2} \approx \frac{3}{n-p}, \tag{41}$$

which approaches zero when $n \gg p$, as desired.

This argument used a somewhat unrealistic kernel eigenspectrum, but numerical experiments show that $C^{(\mathbf{\Phi},\phi)}$ is approximately deterministic even for real eigenspectra (Figure F4). We leave the analysis of $C^{(\mathbf{\Phi},\phi)}$ for realistic kernels for future work.

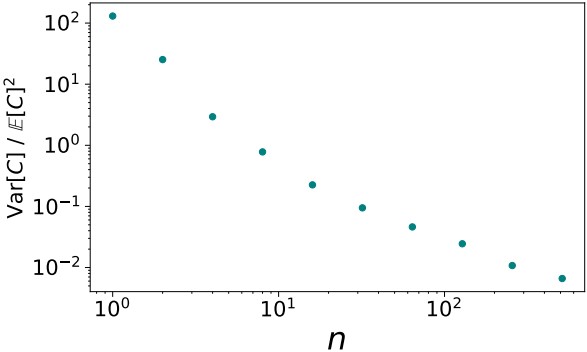

Figure F4: **For NTK eigenvalues on the 7-sphere,** $\mathrm{Var}\left[C^{(\mathbf{\Phi},\phi)}\right] \ll \bar{C}$ **for large** $n$**.** The statistic $\mathrm{Var}\left[C^{(\mathbf{\Phi},\phi)}\right]/\bar{C} \to 0$ is calculated for various $n$ using 100 samples of $\mathbf{\Phi}, \phi$ with the 7-sphere eigenvalues up to $k=6$.

Replacing $C^{(\mathbf{\Phi},\phi)}$ by a constant $C$ yields our final approximation for $\mathbf{T}$, given in Equation 11. We note that, though this constant $C$ is roughly the mean of $C^{(\mathbf{\Phi},\phi)}$, we circumvent deriving an explicit expression for this mean by instead using Theorem 1 to fix $C$.

### F.1 PROOF OF LEMMA 2

Here we provide a proof of the Lemma 2, which states several properties of the constant $C$ defined by Equation 11. As stated in the Lemma, these properties assume $\{\lambda_i\}_{i=1}^{M}$ ordered from greatest to least.

**Property (a)**: $C = \infty$ when $n = 0$, and $C = 0$ when $n = M$.

*Proof.* Because $\sum_{i=1}^{M} \frac{\lambda_i}{\lambda_i + C}$ is strictly decreasing with $C$ for $C \geq 0$, there can only be one solution for a given $n$. The first statement follows by inspection, and the second follows by inspection given our assumption that all eigenvalues are strictly positive.

**Property (b)**: $C$ is strictly decreasing with $n$.

*Proof.* Differentiating the constraint on $C$ with respect to $n$ yields

$$\sum_{i=1}^{M} \frac{-\lambda_i}{(\lambda_i + C)^2} \frac{dC}{dn} = 1, \tag{42}$$

giving

$$\frac{dC}{dn} = -\left[ \sum_{i=1}^{M} \frac{\lambda_i}{(\lambda_i + C)^2} \right]^{-1} < 0. \tag{43}$$

**Property (c)**: $C \leq \frac{1}{n-\ell} \sum_{i=\ell+1}^{M} \lambda_i$ for all $\ell \in \{0, ..., n-1\}$.

*Proof.* $n = \sum_{i=1}^{M} \frac{\lambda_i}{\lambda_i + C} \leq \ell + \sum_{i=\ell+1}^{M} \frac{\lambda_i}{C}$. The desired property follows.

**Property (d)**: $C \geq \lambda_\ell \left( \frac{\ell}{n} - 1 \right)$ for all $\ell \in \{n, ..., M\}$.

*Proof.* We consider replacing $\lambda_i$ with $\lambda_\ell$ if $i \leq \ell$ and $0$ if $i > \ell$. Noting that this does not increase any term in the sum, we find that $n = \sum_{i=1}^{M} \frac{\lambda_i}{\lambda_i + C} \geq \sum_{i=1}^{\ell} \frac{\lambda_\ell}{\lambda_\ell + C} = \frac{\ell \lambda_\ell}{\lambda_\ell + C}$. The desired property follows. $\square$

## G  Second-order statistics of $\mathbf{T}^{(\mathcal{D})}$

### G.1  Preliminaries

Here we derive expressions for the second-order statistics of $\mathbf{T}^{(\mathcal{D})}$. These derivations make no further approximations beyond those already made in approximating $\mathbf{T}$.

We begin a calculation that will later be of use. Differentiating both sides of the constraint on $C$ in Equation 11 with respect to a particular eigenvalue, we find that

$$\frac{d}{d\lambda_i} \sum_{j=1}^{M} \frac{\lambda_j}{\lambda_j + C} = \sum_{j=1}^{M} \frac{-\lambda_j}{(\lambda_j + C)^2} \frac{dC}{d\lambda_i} + \frac{C}{(\lambda_i + C)^2} = 0, \tag{44}$$

yielding that

$$\frac{dC}{d\lambda_i} = \frac{C}{q(\lambda_i + C)^2}, \quad \text{where} \quad q \equiv \sum_{j=1}^{M} \frac{\lambda_j}{(\lambda_j + C)^2}. \tag{45}$$

We now factor $\mathbf{T}^{(\mathcal{D})}$ into two matrices as

$$\mathbf{T}^{(\mathcal{D})} = \mathbf{\Lambda Z}, \quad \text{where} \quad \mathbf{Z} \equiv \mathbf{\Phi} \left( \mathbf{\Phi}^T \mathbf{\Lambda} \mathbf{\Phi} \right)^{-1} \mathbf{\Phi}^T. \tag{46}$$

Unlike $\mathbf{T}^{(\mathcal{D})}$, the matrix $\mathbf{Z}$ has the advantage of being symmetric and containing only one factor of $\mathbf{\Lambda}$. Our approach will be to study the second-order statistics of $\mathbf{Z}$, which will trivially give these statistics for $\mathbf{T}^{(\mathcal{D})}$. From Equation 11, we can approximate the mean of $\mathbf{Z}$ as

$$\mathbb{E}_{\mathbf{\Phi}}[\mathbf{Z}] = (\mathbf{\Lambda} + C\mathbf{I}_M)^{-1}. \tag{47}$$

We also define a modified matrix $\mathbf{Z}^{(\mathbf{U})} \equiv \mathbf{\Phi}\left(\mathbf{\Phi}^T\mathbf{U}^T\mathbf{\Lambda}\mathbf{U}\mathbf{\Phi}\right)^{-1}\mathbf{\Phi}^T$, where $\mathbf{U}$ is an orthogonal $M \times M$ matrix. Because the measure over which $\mathbf{\Phi}$ is averaged is rotation-invariant, we can equivalently average over $\tilde{\mathbf{\Phi}} \equiv \mathbf{U}\mathbf{\Phi}$ with the same measure, giving

$$\mathbb{E}_{\mathbf{\Phi}}\left[\mathbf{Z}^{(\mathbf{U})}\right] = \mathbb{E}_{\tilde{\mathbf{\Phi}}}\left[\mathbf{U}^T\tilde{\mathbf{\Phi}}\left(\tilde{\mathbf{\Phi}}^T\mathbf{\Lambda}\tilde{\mathbf{\Phi}}\right)^{-1}\tilde{\mathbf{\Phi}}^T\mathbf{U}\right] = \mathbb{E}_{\mathbf{\Phi}}\left[\mathbf{U}^T\mathbf{Z}\mathbf{U}\right] = \mathbf{U}^T(\mathbf{\Lambda}+C\mathbf{I}_M)^{-1}\mathbf{U}. \quad (48)$$

It is similarly the case that

$$\mathbb{E}_{\mathbf{\Phi}}\left[(\mathbf{Z}^{(\mathbf{U})})_{ij}(\mathbf{Z}^{(\mathbf{U})})_{k\ell}\right] = \mathbb{E}_{\mathbf{\Phi}}\left[\left(\mathbf{U}^T\mathbf{Z}\mathbf{U}\right)_{ij}\left(\mathbf{U}^T\mathbf{Z}\mathbf{U}\right)_{k\ell}\right]. \quad (49)$$

Our aim will be to calculate expectations of the form $\mathbb{E}_{\mathbf{\Phi}}[\mathbf{Z}_{ij}\mathbf{Z}_{k\ell}]$. With a clever choice of $\mathbf{U}$, we can now see that most choices of the four indices will make this expression zero. We define $\mathbf{U}_{ab}^{(m)} \equiv \delta_{ab}(1-2\delta_{am})$ and observe that, because $\mathbf{\Lambda}$ is diagonal, $(\mathbf{U}^{(m)})^T\mathbf{\Lambda}\mathbf{U}^{(m)} = \mathbf{\Lambda}$ and thus $\mathbf{Z}^{(\mathbf{U}^{(m)})} = \mathbf{Z}$. Equation 49 then yields that

$$\mathbb{E}_{\mathbf{\Phi}}[\mathbf{Z}_{ij}\mathbf{Z}_{k\ell}] = (-1)^{\delta_{im}+\delta_{jm}+\delta_{km}+\delta_{\ell m}}\mathbb{E}_{\mathbf{\Phi}}[\mathbf{Z}_{ij}\mathbf{Z}_{k\ell}], \quad (50)$$

from which it follows that $\mathbb{E}_{\mathbf{\Phi}}[\mathbf{Z}_{ij}\mathbf{Z}_{k\ell}] = 0$ if any index is repeated an odd number of times. In light of the fact that $\mathbf{Z}_{ij} = \mathbf{Z}_{ji}$, there are only three distinct nontrivial cases to consider:

1. $\mathbb{E}_{\mathbf{\Phi}}[\mathbf{Z}_{ii}\mathbf{Z}_{ii}]$,
2. $\mathbb{E}_{\mathbf{\Phi}}[\mathbf{Z}_{ij}\mathbf{Z}_{ij}]$ with $i \neq j$, and
3. $\mathbb{E}_{\mathbf{\Phi}}[\mathbf{Z}_{ii}\mathbf{Z}_{jj}]$ with $i \neq j$.

We stress that we are not using the Einstein convention of summation over repeated indices.

## G.2 CASE STUDIES

**Cases 1 and 2.** We now consider differentiating $\mathbf{Z}$ with respect to a particular element of the matrix $\mathbf{\Lambda}$. This yields

$$\frac{d\mathbf{Z}_{i\ell}}{d\mathbf{\Lambda}_{jk}} = -\phi_i^T\left(\mathbf{\Phi}^T\mathbf{\Lambda}\mathbf{\Phi}\right)^{-1}\phi_j\phi_k^T\left(\mathbf{\Phi}^T\mathbf{\Lambda}\mathbf{\Phi}\right)^{-1}\phi_\ell = -\mathbf{Z}_{ij}\mathbf{Z}_{k\ell}, \quad (51)$$

where $\phi_i$ is the $i$-th row of $\mathbf{\Phi}$. This gives us the useful expression that

$$\mathbb{E}_{\mathbf{\Phi}}[\mathbf{Z}_{ij}\mathbf{Z}_{k\ell}] = -\frac{d}{d\mathbf{\Lambda}_{jk}}\mathbb{E}_{\mathbf{\Phi}}[\mathbf{Z}]. \quad (52)$$

We now set $\ell = i$ and evaluate this expression using Equation 47, concluding that

$$\mathbb{E}_{\mathbf{\Phi}}[\mathbf{Z}_{ij}\mathbf{Z}_{ij}] = \mathbb{E}_{\mathbf{\Phi}}[\mathbf{Z}_{ij}\mathbf{Z}_{ji}] = -\frac{d}{d\lambda_j}\left(\frac{1}{\lambda_i+C}\right) = \frac{1}{(\lambda_i+C)^2}\left(\delta_{ij}+\frac{C}{q(\lambda_j+C)^2}\right), \quad (53)$$

$$\mathrm{Cov}_{\mathbf{\Phi}}[\mathbf{Z}_{ij},\mathbf{Z}_{ij}] = \mathrm{Cov}_{\mathbf{\Phi}}[\mathbf{Z}_{ij},\mathbf{Z}_{ji}] = \frac{C}{q(\lambda_i+C)^2(\lambda_j+C)^2}. \quad (54)$$

We did not require that $i \neq j$, and so Equation 53 holds for Case 1 as well as Case 2.

**Case 3.** We now aim to calculate $\mathbb{E}_{\mathbf{\Phi}}[\mathbf{Z}_{ii}\mathbf{Z}_{jj}]$ with $i \neq j$. We might hope to use Equation 52 in calculating $\mathbb{E}_{\mathbf{\Phi}}[\mathbf{Z}_{ii}\mathbf{Z}_{jj}]$, but this approach is stymied by the fact that we would need to take a derivative with respect to $\mathbf{\Lambda}_{ij}$ we only have an approximation for $\mathbf{Z}$ for diagonal $\mathbf{\Lambda}$. We can circumvent this by means of $\mathbf{Z}^{(\mathbf{U})}$. From the definition of $\mathbf{Z}^{(\mathbf{U})}$, we find that

$$\left(\frac{d}{d\mathbf{U}_{ij}} - \frac{d}{d\mathbf{U}_{ji}}\right)\mathbf{Z}^{(\mathbf{U})}\bigg|_{\mathbf{U}=\mathbf{I}_M}$$

$$= -\boldsymbol{\phi}_i^T \left(\boldsymbol{\Phi}^T\boldsymbol{\Lambda}\boldsymbol{\Phi}\right)^{-1}\left[\boldsymbol{\phi}_j\lambda_i\boldsymbol{\phi}_i^T - \boldsymbol{\phi}_i\lambda_j\boldsymbol{\phi}_j^T + \boldsymbol{\phi}_i\lambda_i\boldsymbol{\phi}_j^T - \boldsymbol{\phi}_j\lambda_j\boldsymbol{\phi}_i^T\right]\left(\boldsymbol{\Phi}^T\boldsymbol{\Lambda}\boldsymbol{\Phi}\right)^{-1}\boldsymbol{\phi}_j$$

$$= (\lambda_j - \lambda_i)\left(\mathbf{Z}_{ij}^2 + \mathbf{Z}_{ii}\mathbf{Z}_{jj}\right). \quad (55)$$

Differentiating with respect to both $\mathbf{U}_{ij}$ and $\mathbf{U}_{ji}$ with opposite signs ensures that the derivative is taken within the manifold of orthogonal matrices. Now, using Equation 48, we find that

$$\left(\frac{d}{d\mathbf{U}_{ij}} - \frac{d}{d\mathbf{U}_{ji}}\right)\mathbb{E}_{\boldsymbol{\Phi}}\left[\mathbf{Z}^{(\mathbf{U})}\right]\bigg|_{\mathbf{U}=\mathbf{I}_M} = \left(\frac{d}{d\mathbf{U}_{ij}} - \frac{d}{d\mathbf{U}_{ji}}\right)\mathbf{U}^T(\boldsymbol{\Lambda} + C\mathbf{I}_M)^{-1}\mathbf{U}\bigg|_{\mathbf{U}=\mathbf{I}_M}$$

$$= \frac{1}{\lambda_i + C} - \frac{1}{\lambda_j + C}. \quad (56)$$

Taking the expectation of Equations 55, plugging in Equation 53 for the squared off-diagonal element, comparing to 56, and performing a bit of algebra, we conclude that

$$\mathbb{E}_{\boldsymbol{\Phi}}[\mathbf{Z}_{ii}\mathbf{Z}_{jj}] = \frac{1}{(\lambda_i + C)(\lambda_j + C)} - \frac{C}{q(\lambda_i + C)^2(\lambda_j + C)^2} \quad (57)$$

and that $\mathbf{Z}_{ii}, \mathbf{Z}_{jj}$ are anticorrelated with covariance

$$\mathrm{Cov}_{\boldsymbol{\Phi}}[\mathbf{Z}_{ii}, \mathbf{Z}_{jj}] = -\frac{C}{q(\lambda_i + C)^2(\lambda_j + C)^2}. \quad (58)$$

With the use of Kronecker deltas, we can combine Equations 54 and 58 into one expression covering all cases. As can be verified by case-by-case evaluation, one such expression is

$$\mathrm{Cov}_{\boldsymbol{\Phi}}[\mathbf{Z}_{ij}, \mathbf{Z}_{k\ell}] = \frac{C\left(\delta_{ik}\delta_{j\ell} + \delta_{i\ell}\delta_{jk} - \delta_{ij}\delta_{k\ell}\right)}{q(\lambda_i + C)(\lambda_j + C)(\lambda_k + C)(\lambda_\ell + C)}. \quad (59)$$

Using the fact that $\mathbf{T}_{ij}^{(\mathcal{D})} = \lambda_i\mathbf{Z}_{ij}$, we obtain the elementwise covariances of $\mathbf{T}^{(\mathcal{D})}$ given in Equation 13.

### G.3 DERIVING AN EXPRESSION FOR MSE

Mean MSE is given by $\mathcal{E}(f) = \mathbb{E}\left[(\hat{\mathbf{v}} - \mathbf{v})^2\right] = \mathbf{v}^T\mathbb{E}\left[(\mathbf{T}^{(\mathcal{D})} - \mathbf{I}_M)^T(\mathbf{T}^{(\mathcal{D})} - \mathbf{I}_M)\right]\mathbf{v} = \mathbf{v}^T\mathbf{E}\mathbf{v}$, where $\mathbf{v}$ and $\hat{\mathbf{v}}$ are the eigenfunction coefficients of the true and learned functions and we have defined $\mathbf{E} \equiv \mathbb{E}\left[(\mathbf{T}^{(\mathcal{D})} - \mathbf{I}_M)^T(\mathbf{T}^{(\mathcal{D})} - \mathbf{I}_M)\right]$. We can derive Equation 15 as follows:

$$
\begin{aligned}
\mathbf{E}_{ij} &= \mathbb{E}\left[\sum_{k=1}^{M}\left(\mathbf{T}_{ki}^{(\mathcal{D})} - \delta_{ik}\right)\left(\mathbf{T}_{kj}^{(\mathcal{D})} - \delta_{jk}\right)\right] \\
&= \sum_{k=1}^{M}\mathbb{E}\left[\mathbf{T}_{ki}^{(\mathcal{D})}\mathbf{T}_{kj}^{(\mathcal{D})}\right] - \mathbb{E}\left[\mathbf{T}_{ji}^{(\mathcal{D})}\right] - \mathbb{E}\left[\mathbf{T}_{ij}^{(\mathcal{D})}\right] + \delta_{ij} \\
&= \delta_{ij}\left[\sum_{k=1}^{M}\frac{\lambda_k^2}{(\lambda_k + C)^2}\left(\delta_{ik} + \frac{C}{q(\lambda_i + C)^2}\right) - 2\frac{\lambda_i}{\lambda_i + C} + 1\right] \\
&= \delta_{ij}\left[\left(1 - \frac{\lambda_i}{\lambda_i + C}\right)^2 + \frac{C}{q(\lambda_i + C)^2}\sum_{k=1}^{M}\left(\frac{\lambda_k}{\lambda_k + C} - \frac{C\lambda_k}{(\lambda_k + C)^2}\right)\right] \\
&= \delta_{ij}\left[\left(\frac{C}{\lambda_i + C}\right)^2 + \frac{nC}{q(\lambda_i + C)^2} - \frac{qC^2}{q(\lambda_i + C)^2}\right] \\
&= \frac{\delta_{ij}nC}{q(\lambda_i + C)^2}.
\end{aligned}
\tag{60}
$$

Though it is not apparent at a glance, this expression for MSE is in fact equivalent to the main result of Bordelon et al. (2020) with a ridge parameter of zero. To obtain their expression from ours, substitute $n \to p$, $C \to t/p$, $q \to p^2/t - p^3\gamma/t^3$, and $v_i \to \lambda_i^{1/2}w_i$. Given that our derivations are quite different — they provide one using the method of characteristics and one using the replica trick from statistical mechanics that both directly yield final approximate expressions, while ours first derives an approximation with one unknown parameter $C$ and then later determines $C$ with the no-free-lunch theorem — it is interesting that we nonetheless recover their results. We note that our results for other second-order statistics besides MSE did not appear in this or any other prior work.

### G.4 SECOND-ORDER STATISTICS CAN BE WRITTEN STRICTLY IN TERMS OF LEARNABILITY

As a final theoretical note, we observe that our expressions for transfer matrix covariances and MSE can all be written strictly in terms of the modewise learnabilities $\mathcal{L}_i \equiv \frac{\lambda_i}{\lambda_i + C}$. The fact that this is possible supports our claim that modewise learnabilities are quantities of fundamental theoretical importance to kernel regression. We begin by noting that

$$
n = \sum_{m=1}^{M}\mathcal{L}_m \quad \text{and} \quad Cq = \sum_{m=1}^{M}\mathcal{L}_m(1 - \mathcal{L}_m).
\tag{61}
$$

With these expressions in hand, we can quickly see that

$$
\text{Cov}\left[\mathbf{T}_{ij}^{(\mathcal{D})}, \mathbf{T}_{k\ell}^{(\mathcal{D})}\right] = \frac{\mathcal{L}_i(1 - \mathcal{L}_j)\mathcal{L}_k(1 - \mathcal{L}_\ell)}{\sum_{m=1}^{M}\mathcal{L}_m(1 - \mathcal{L}_m)}(\delta_{ik}\delta_{j\ell} + \delta_{i\ell}\delta_{jk} - \delta_{ij}\delta_{k\ell})
\tag{62}
$$

and

$$
\mathbf{E}_{ii} = \frac{\sum_{m=1}^{M}\mathcal{L}_m}{\sum_{m'=1}^{M}\mathcal{L}_{m'}(1 - \mathcal{L}_{m'})}(1 - \mathcal{L}_i)^2.
\tag{63}
$$

These expressions allow one to more clearly see the effects of the model's inductive bias. For example, in light of the fact that $\mathcal{L}_i \in [0,1]$, Equation 62 shows that $\text{Var}\left[\mathbf{T}_{ij}^{(\mathcal{D})}\right]$ is larger the higher the learnability of the "learned mode" $i$ and the *lower* the learnability of the "target mode" $j$, meaning that low-learnability modes tend to be mistaken for high-learnability modes. Since the model tends to err towards high-learnability modes, we thus expect that the learned function $\hat{f}$ will tend to be spectrally biased, placing outsize weight in more learnable modes.

One recurring quantity in these expressions is $\mathcal{L}_i(1 - \mathcal{L}_i)$, which is zero for $\mathcal{L}_i = 0, 1$ and maximal at $\mathcal{L}_i = .5$. Intuitively speaking, it describes the rate at which a mode is "currently being learned" with increasing $n$, as can be seen from the fact that

$$\frac{d\mathcal{L}_i}{dn} = \frac{\mathcal{L}_i(1 - \mathcal{L}_i)}{\sum_{m=1}^{M} \mathcal{L}_m(1 - \mathcal{L}_m)}. \tag{64}$$

This equation states that a new unit of learnability (i.e. a new data point) is divvied up among all the eigenmodes in proportion to $\mathcal{L}_i(1 - \mathcal{L}_i)$.

## H    EXPERIMENTAL DETAILS

We conduct all our experiments using JAX, performing exact NTK regression with the Neural Tangents library (Novak et al., 2019) built atop it. For the dataset sizes we consider in this paper, exact NTK regression is typically quite fast, running in seconds, while the training time of finite networks varies from seconds to minutes and depends on width, depth, training set size, and eigenmode. In particular, as described by Rahaman et al. (2019), lower eigenmodes take longer to train (especially when aiming for near-zero training MSE as we do here).

Unless otherwise stated, all experiments used four-hidden-layer ReLU networks initialized with NTK parameterization (Sohl-Dickstein et al., 2020) with $\sigma_w = 1.4$, $\sigma_b = .1$. The $\tanh$ networks used in generating Figure 3 instead used $\sigma_w = 1.5$. Experiments on the unit circle always used a learning rate of .5, while experiments on the hypercube and hypersphere used a learning rate of .5 or .1 depending on the experiment. While higher learning rates led to faster convergence, they often also gave qualitatively different generalization behavior, in line with the large learning rate regimes described by Lewkowycz et al. (2020). Means and $1\sigma$ error bars always reflect statistics from 30 random dataset draws and initializations (for finite nets), except for the nonmonotonic MSE curves of Figure A1, which used 100 trials. We emphasize that the "no free lunch" experiment of Figure 3 used only a single trial.

# I  NTK EIGENVALUES ON EXPERIMENTAL DOMAINS

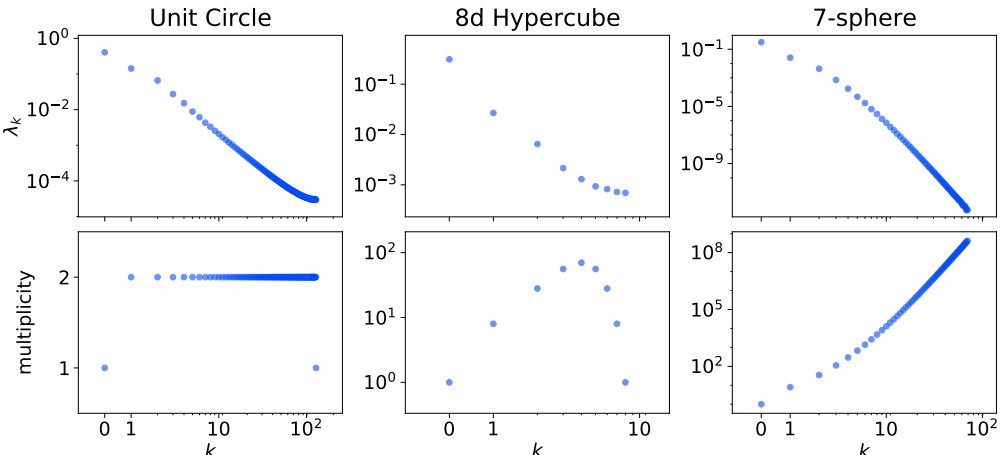

Figure I5: **Eigenvalues decrease with increasing $k$ for all three domains, the manifestation of an inductive bias towards smoother functions.** All eigenvalues are calculated with the NTK of a 4L ReLU network as described in the text. **(A)** NTK eigenvalues for $k$ for the discretized unit circle ($M = 256$). Eigenvalues decrease as $k$ increases except for a few near exceptions at high $k$. **(B)** NTK eigenvalues for the 8d hypercube. Eigenvalues decrease monotonically with $k$. **(C)** NTK eigenvalues for the 7-sphere up to $k = 70$. Eigenvalues decrease monotonically with $k$. **(D)** Eigenvalue multiplicity for the discretized unit circle. All eigenvalues are doubly degenerate (due to $cos$ and $sin$ modes) except for $k = 0$ and $k = 128$. **(E)** Eigenvalue multiplicity for the 8d hypercube. **(F)** Eigenvalue multiplicity for the 7-sphere.

