# OpenReview forum: "Neural tangent kernel eigenvalues accurately predict generalization"
_ICLR.cc/2022/Conference — ICLR 2022 Submitted_

### Official Review · Reviewer_q2g8 · 2021-10-28

**Correctness:** 3
**Technical Novelty And Significance:** 4
**Empirical Novelty And Significance:** 1
**Recommendation:** 8
**Confidence:** 5

**Main Review:**

This paper provides a really nice theoretical analysis of generalization in the setting of kernel regression. The metric of "learnability" is intuitive *and* useful, and the new no-free-lunch theorem provides new insight into the relationship between the size of the training set and the success of regression. In particular, it explains how the inductive bias of the kernel relates to the properties of the function that is trying to be learned.

I think a major weakness of the paper is the desire to extend the regime of these results to make claims about deep learning. In particular, deep neural neural networks are only kernel machines at infinite width, and we expect deviations from this limit for realistic finite width networks. It's well known that the kernel for finite width networks is not fixed and evolves during training, so it's rather hard to assess the relevance of the paper's results for the models that the authors really want their results to apply to (namely, neural network models). In addition, for these finite width networks, the eigenspectrum also would shift, and thus the notion of which modes are large may also become complicated.

Relatedly, the specific experiments performed are on very artificial datasets. That certainly can be fine -- the results illustrate the validity of the kernel regression results(!) -- but it also gives me pause with the claims made about deep learning in general. One particular problem is that the spectrum of eigenvalues for realistic data often obeys a power law. (See e.g. some discussion in [Explaining Neural Scaling Laws](https://arxiv.org/abs/2102.06701), or just check it explicitly on MNIST or FMNIST with a few lines of code.) This seems to be an essential inductive bias of realistic datasets. Presumably, given the no-free-lunch theorem, to make claims about realistic deep learning scenarios we'd need to understand (a) what an eigendecomposition of typical functions we want to learn looks like and (b) how kernel regression solutions behave when the data has a power law spectrum. I suspect that there are important differences as compared to learning an eigenfunction on an artificial dataset. Along these lines, it seems that the most relevant question is why is the inductive bias of fully-connected networks, which could be assessed by looking at the particular combinations of parameters and activations that make up the NTK, often useful for learning and generalizing the datasets and functions that we typically care about for AI function approximation tasks.

Finally, I am particularly bothered by the discussion finite width networks in the abstract, introduction, and final paragraph before the conclusion. Since finite-width neural networks are not kernel machines,  the first-principles theory of this paper can only approximately apply to them. I think the experiments at finite width in this paper -- primarily in Figure B2 of the supplement -- are not sufficient to support the various claims made about finite-width networks. First, as described above, it's not clear that the learning tasks are particularly realistic. Second, it's now pretty well known that the relevant quantity with which to compare the width of the network to assess the validity of the infinite-width limit is the depth of the network. Thus, simply stating that their experiments agree for "networks as narrow as width 20" is not particularly helpful or scientifically accurate. A width 20 network of 1 layer might be very much like kernel regression and be in a regime of applicability of their results, but a width 20 network of 20 layers will definitely not be. Of course, the relevant question is whether for typical width and depth combinations whether their results are still useful. But I think this is really hard to assess given the experiments are also on unrealistic data and learning tasks.

One minor comment: I wish the authors would compare their results more directly to the analysis of the Canatar et al. 2021 paper they cite. I understand that the no-free-lunch theorem, the learnability measure, and the analysis of the learning transfer matrix are new here, but both works purport to give a first-principles theory of generalization by studying the kernel eigenspectrum, and I would appreciate a deeper discussion of the connection.

A tiny comment: It would be nice to have more discussion of what Theorem 2 means after giving it. (But I know that space constraints makes things difficult.)

**Summary Of The Paper:**

This paper provides a novel theoretical account of generalization for kernel regression. To do so, the authors study a matrix built out of the kernel eigensystem evaluated on the training set that they call the "learning transfer matrix," and which relates the decomposition of the true function in the eigenbasis to the decomposition of the learned function in the eigenbasis. In other words, this matrix characterizes the kernel regression solution in the eigenbasis. By making a number of approximations, they are able to find a closed-form expression for this matrix and study its first and second order statistics. A main conclusion from this analysis is that functions are more learnable by kernel regression if they have more weight in higher eigenvalue modes.

Moreover, the authors also introduce a new metric, which they call "learnability," that they use to prove a new no-free-lunch theorem whose content is that when averaging over a complete basis of functions the learnability is independent of the kernel. This means that the choice of kernel should be tailored to the details of the function being learned in order for kernel regression to succeed. A related result is that there are some functions for which the kernel regression solution generalizes worse than simply outputting "0" on data outside the training set, i.e. for which the solution fails to generalize at all.


**Summary Of The Review:**

On the one hand, this is a really nice theoretical paper about generalization for kernel regression. I think the results are useful, intuitive, correct, and bring new insight that let us understand better how kernel regression works. On the other hand, I do not think the analysis supports that this is a first-principles theory of generalization in deep learning. The results may very well be relevant for explaining generalization in deep neural networks, but the theoretical analysis and experimental evidence for that are not particularly convincing as of yet.

My recommendation that this is marginally below the acceptance threshold is based on the framing of the paper and statements that about deep learning that are not as careful as the really nice analysis of kernel regression is. If some of these statements are revised (or perhaps supported with more evidence), and/or if there's more discussion of when they expect their results to apply to DNNs and when they expect it to break down, I will substantially increase my score and recommend an acceptance.

### After Author Responses

The authors have addressed a number of my concerns, and I have updated my score accordingly.

---

> ### Author Response · Authors · 2021-11-13
> **Summary of changes in response to reviewer comments (1/2)**
>
> Thank you for the careful review. We are gratified that you see appreciable novelty and significance in our new theoretical treatment of kernel regression and particularly in our measure of learnability, our no-free-lunch and worse-than-chance-generalization theorems, and our simple, closed-form expressions for quantities of interest. We feel you understand our central results and are correct in your assessment that they constitute significant new insight into kernel regression.
>
> Regarding your central criticism, we certainly do not claim to have derived any results for finite networks, whose NTKs both depend on initialization and evolve throughout training. These changes in the NTK are small if width is large (scaling as $\mathcal{O}(n^{-1/2})$), so one should expect our results to approximately hold for wide networks, but the theory of this paper treats only exact kernel regression. Rereading our paper, however, we can see how certain statements could be interpreted as general claims regarding finite networks, and we agree that these should be disambiguated. We have therefore rephrased many sentences, particularly in our abstract, introduction, and conclusion, to clarify that our paper gives a theoretical treatment of *kernel regression*, and extensions to neural networks are only exact in the infinite-width limit. We do claim, however, that the excellent empirical agreement between our theory's predictions and width-500, depth-4 networks suggests our theory is a promising starting point for developing a theory of generalization applicable to finite networks.
>
> Regarding the synthetic nature of our learning problems, while we emphasize that our exact results hold for kernel regression regardless of the dataset, we agree that it is a limitation of our empirical work that we treat only artificial problems. This is a consequence of the need for knowledge of the eigensystem in order to make and verify our theory's predictions: very little is known about the eigensystems (or even the distributions) of natural image data besides the power law NTK spectra you note. On that point, however, we note that *our unit circle dataset indeed has a power law spectrum with exponent ~1.5*, as shown in Fig. I5. Furthermore, since kernel regression is linear in $f$, if our theory is accurate for all *eigenfunctions* on this dataset (as our experiments suggest it is), then it is accurate for all functions on the dataset. Our unit circle experiments therefore suggest that our theory will remain accurate even for complex target functions on domains with power-law eigenspectra. We also note that one workaround for this problem (used by Canatar et al.) is to let $\mathcal{X}$ be the train + test set of, e.g., MNIST; we did not use this trick because we felt it was potentially confusing to readers that generating predictions requires knowledge of the test dataset. We will note the unit circle power-law spectrum in the final version of the paper.
>
> Regarding your note about narrow architectures, we agree that our results are not definitive and that we should have noted the depth of these networks. Accordingly, we have modified each of the statements you mentioned to mention the depth and to attenuate their claims. Most strongly, in the Experiments section, we have changed "This surprising agreement suggests our theory will faithfully predict generalization in practical deep learning systems" to "This surprising agreement suggests our theory is a promising starting point for understanding the generalization of practical neural networks." We thank you for noting our too-strong claim.

---

> > ### Author Response · Authors · 2021-11-13
> > **Summary of changes in response to reviewer comments (2/2)**
> >
> > With respect to your desire for more explicit comparison with Canatar et al, in response to all reviewers' feedback, we will be adding a dedicated "Related Works" section to our paper clarifying its connections to and improvements over prior works including this paper. This will appear soon in a revised PDF. For your immediate review, though, here are the main distinctions between our work and Bordelon et al. and Canatar et al. (which have the same authors); please let us know if any require further clarification.
> > * Their main results all rely on non-rigorous approximations. By contrast, we prove many results exactly.
> > * Our measure of learnability is new, as is our no-free-lunch theorem. We feel this new "conservation law" perspective, in which designing a kernel amounts to allocating its $n$ units of learnability, is very valuable both theoretically and practically. For example, if one wants to design the best kernel for a task, it will be much easier to work in terms of learnabilities than in terms of eigenvalues.
> > * Our theorem regarding worse-than-chance generalization is new.
> > * The experiments of these works only study MSE, while we also study a few other functionals of $\hat{f}$. MSE is not the only important quantity; for example, if one wanted to study adversarial examples, one might want to know about the expectation of $|\nabla_x \hat{f}(x)|^2$, which could be written as a 2nd-order statistic of $\hat{f}$ and analytically predicted with our theory.
> > * Our derivation is quite different, even when it arrives at the same results. They use a lengthy but direct replica method calculation, while we use a much simpler method to find the functional form of learnability, then solve for $C$ using the no-free-lunch theorem. We feel that our derivation is, in many respects, easier to understand, and regardless it is quite interesting that two very different methods give the same result (both we and these authors find this intriguing).
> > * Our observation and explanation of nonmonotonic loss curves at low $n$ is new. Actually, as far as we know, this is a qualitatively new phenomenon: we know of no other works, empirical or theoretical, that note that MSE can, e.g., be worse at one data point than at zero. Of course our explanation only applies to kernel regression, but we empirically observe this in neural networks, too. We note that, while theory papers aim to explain findings of experimental papers, here we do the opposite: we theoretically predict a surprising new effect and then go on to observe it in experiment. Due to the lack of fundamental theory, there are very few instances in deep learning theory of this! We would have stressed this point in the paper were it not for space constraints. Further, we don't show it in this paper, but we actually see this even for MNIST and CIFAR10, so this is not purely a result of the synthetic datasets.
> >
> > Thank you again for your review; we would be happy to make further changes if you have any to request. We hope our revision appreciably affects your judgment of our paper.

---

### Official Review · Reviewer_Mosm · 2021-11-02

**Correctness:** 3
**Technical Novelty And Significance:** 2
**Empirical Novelty And Significance:** 3
**Recommendation:** 5
**Confidence:** 3

**Main Review:**

- Section 2.1: In the opening part there is no assumption about independence of outputs (m of them). Judging by the claim on page 3 (the last paragraph), the paper does assume outputs are independent? Otherwise, I fail to see how the problem simplifies to m = 1. To see that these kernel-based problems are quite different, please check out "On learning vector-valued functions" by Micchelli and Pontil (2003).
- Figure of merit: The inner product in Eq. (2) is cosine similarity and should be in [-1, 1]. Unless one takes the absolute value, I fail to see how this quantity is bounded in [0, 1]?
- Section 2.3: This section appears to be a review of kernel methods rather than a novel theoretical contribution. In particular, it does not appear to depend on any assumptions or specifics of neural tangent kernels. If so, then this should be contrasted relative to prior work on kernels and eigendecomposition of kernel operators. For example, "On the mathematical foundations of learning" by Cucker & Smale (2002) or "Reproducing kernel Hilbert spaces in Probability and Statistics" by Berlinet & Thomas-Agnan.
- Section 2.5: The discussion immediately after Lemma 2. This is about the dependence on the sample size and, thus, not sure why this is specific to neural tangent kernel? In kernel methods the hypothesis will concentrate around the target as the sample size increase. There are numerous bounds on this and prior work has also covered the spectral tail, interpolation, etc (see "Just Interpolate: Kernel Ridgeless Regression Can Generalize" by Liang & Rakhlin or "Optimal Rates for the Regularized Least-Squares Algorithm" by Caponnetto & de Vito).

The empirical analysis is interesting and should have been the main focus of the work. In this regard, the ablation study with input spaces that have the closed form expressions for eigenfunctions is quite informative. The observation that even for networks with only 20-unit widths the correlations persist is promising.

**Summary Of The Paper:**

The paper aims at characterizing the generalization ability of a neural network via the spectral properties of the neural tangent kernel.

**Summary Of The Review:**

The paper should have focused on its strengths which is the empirical study that links the NTK spectrum to generalization in neural networks. The theoretical results apply to kernels in general and, thus, do not see much novelty in this regard. The related work does not adequately cover prior developments in kernel methods and fails to place the claimed theoretical contributions within this scope.
I would recommend a major revision with a proper review of prior work on kernels. The focus of the paper should be empirical and presentation of its empirical findings should receive more attention/pages. I also fail to see sufficient theoretical contribution relative to prior work on kernel methods.

---

> ### Author Response · Authors · 2021-11-13
> **Response to reviewer comments (1/2)**
>
> Thank you for the review. We are glad that you found its experiments strong and interesting, particularly those at narrow width. However, we strongly disagree that this work does not include novel, significant theoretical results, and we feel you perhaps missed significant contributions that would have affected your evaluation. This is likely due in large part to the fact that we did not make these key contributions significantly clear, and so we will soon add an explicit "Contributions" section to the paper in a revision to the PDF. For your review, here is the theoretical significance of our work; please let us know if anything is unclear or unconvincing.
>
> * Firstly, you are correct that our theory applies not just to NTK regression but to **all** kernel regression (i.e. RBF kernel regression, Laplace kernel regression, linear regression, etc.). We do not understand your critique that our results "apply to kernels in general and, thus, [there is not] much novelty in this regard" - we have indeed proven new results for kernel regression *in general*, and that makes our results far *more* significant, not less! We have rephrased statements in the abstract and introduction to make this clear.
> * Our no-free-lunch theorem (Theorem 1) characterizes a fundamental tradeoff in the inductive bias of kernel regression. It is an extremely simple rule exactly obeyed by any kernel, and thus is likely to be a significant result for future studies of kernel regression. We have not positioned it relative to older kernel literature because we know of nothing comparable; the most relevant prior work is the famous general no-free-lunch theorem from Wolpert, of which our result is a more powerful version specialized to kernel regression.
> * Our worse-than-chance generalization theorem (Theorem 2) is a surprising result that clearly illustrates the usefulness of Theorem 1. We would ask the reviewer if he/she knows of other results showing that, for any infinite-width network (or for kernel regression), there always exist functions on which generalization is worse than chance (!!). If so, we would love to cite them as relevant background. If not, then we are unsure why the reviewer finds this result non-novel and insignificant.
> * In current deep learning theory, there are very few quantities which we can accurately predict from first principles. Our paper derives several new ones. In every experimental figure, when we show that theoretical curves agree with experimental quantities for wide finite networks, this is a validation of a theory that provides *accurate expressions for the generalization behavior of neural networks*. The fact that we provide simple theoretical expressions for quantities of interest makes our results highly significant: without the theoretical expressions in this paper (e.g. Eqns. 12, 13), the field simply has no expressions for these quantities of interest.

---

> > ### Author Response · Authors · 2021-11-13
> > **Response to reviewer comments (2/2)**
> >
> > We now address the specific points raised in the review.
> >
> > Regarding the restriction to $m=1$, we agree that that passage was unclear and have clarified it. The word "independent" was unfortunate - we did not mean to imply any statistical independence, merely that kernel regression by its functional form predicts each target index as if it were a separate function. Here is the revised text:
> > * Examining Equation 1, one finds that the $m$ indices of $f$ can each be treated separately: the learned $\hat{f}$ is equivalent to simply performing kernel regression with each of the $m$ indices as a scalar target function and then vectorizing the results. For simplicity, then, we hereafter assume $m=1$.
> >
> > To see this, it is sufficient to note that $\hat{f}(x) = A f(\mathcal{D})$, where $\hat{f}(x) \in \mathbb{R}^{1 \times m}$ is the learned function, $A \in \mathbb{R}^{1 \times n}$ is the product of two kernel matrices (see Eq. 1), and $f(x) \in \mathbb{R}^{n \times m}$ is the target function on the dataset. Because this is standard matrix multiplication, computing results for one column of $f(x)$ at a time gives the same result as computing the full matrix multiplication.
> >
> > Regarding Eq. 2, we agree this was unclear: the fact that it is bounded in [0,1] is proven in Lemma 1d. We have noted this in the paper. (We also note that Eq. 2 is not cosine similarity; the denominator is different.)
> >
> > Section 2.3 does indeed begin with background on standard kernel eigendecomposition; we have added a textbook citation (Shawe-Taylor and Cristianini) to make this clear. However, our formulation of the "learning transfer matrix" *is* novel and proves quite useful, as noted by Reviewer q2g8. We emphasize that our results apply to any kernel and are thus **more** significant and powerful than they would be were they specific to NTKs, not less.
> >
> > Regarding your comment on Section 2.5, again, our results apply to general kernels and are not specific to the NTK. Thank you for bringing these prior works to our attention. Our results are a significant improvement over these - our theory predicts for **all $n$,** not just large $n$ - but they are certainly relevant, and we will discuss them in the introduction of the final paper.
> >
> > Thank you again for your review. We hope that our response has clarified the main points of our paper and highlighted our several novel and significant results. If you have further questions or there is more information we could provide that might help explain why our results are important and useful, please let us know.

---

### Official Review · Reviewer_7tiq · 2021-11-03

**Correctness:** 3
**Technical Novelty And Significance:** 3
**Empirical Novelty And Significance:** 3
**Recommendation:** 6
**Confidence:** 2

**Main Review:**

The topic of the paper which is on the NTK approach of analyzing generalization error of neural networks is interesting. The paper continues the line of work of Bordelon et al 2020 which seems a promising direction. Moreover, the paper is well structured and has plenty of simulation results. However, on the theoretical aspect, I believe the paper can be improved.

I especially found the problem setup section of the paper and its notation hard to read. I think the theoretical results of the paper is a bit incremental with respect to the Bordelon et al. 2020 and Canatar et al. 2021. In particular, Theorems 1 and 2 demonstrate simple facts and have very short proofs. Moreover, the setup of the paper is for finite set $\mathcal{X}$ with $M=|\mathcal{X}|<\infty$ with data sampled from a uniform measure. Although in page 2 it is promised that for non-uniform measures and continuous sets $\mathcal{X}$ results will later be explained, no such analytical explanation is provided except for a note in the simulations section in page 8. The justification for the reduction of output dimension to m=1 in page 3 is also not entirely clear.

In the conclusion section, it should be noted that the results of the paper concern the generalization property of neural networks in the “infinite width” regime, and that the paper merely conjectures that the same theory holds for finite width networks as well.

**Summary Of The Paper:**

The paper examines the eigenvalues of a neural network’s “Neural Tangent Kernel” to analyze its generalization performance in the infinite-width regime. It conjectures that the same results will also apply in the finite width regime as well. By analyzing kernel regression and by defining a measure as the “learnability” of a given target function, the paper proves a “no-free-lunch” theorem which implies that improving a network’s or kernel's generalization for a given target function must worsen its generalization performance for its "orthogonal functions". The paper then analytically predicts two phenomena: worse than chance generalization for hard functions and non-monotonic error curves in a small data regime. It also provides some simulations to corroborate the analytic results.

**Summary Of The Review:**

I am not very familiar with the recent prior work that this paper cites and builds upon its approach and I have not completely checked the mathematical correctness of the claims of the paper. But, overall, given the importance of the problem analyzed by the paper, i.e. generalization performance of neural networks using NTK approach, and due to mix of theoretical results of the paper with empirical results, I am inclined towards accepting the paper.

---

> ### Author Response · Authors · 2021-11-13
> **Response to reviewer comments**
>
> Thank you for the review. We are glad that you find our work well-motivated and well-written and our empirical results thorough. We are happy to see that you have given us an accepting score of 6. However, we feel that this review overlooks significant theoretical contributions beyond those of Bordelon et al. and Canatar et al., and that this recommendation would perhaps be much more favorable were they fully understood. To this end, we will soon add a dedicated "Related Works" section to our paper in which we make explicit our work's advances over prior papers including these. For your review, here are our principal advances over these two works:
>
> * Their main results all rely on non-rigorous approximations, even for exact kernel regression. By contrast, we prove many results exactly, and so we can have confidence that they are true regardless of the kernel and dataset.
> * Our measure of learnability is new, as is our no-free-lunch theorem. As noted by the last reviewer, this no-free-lunch theorem is a simple, novel result giving insight into the inductive bias of *all* forms of kernel regression. This zero-sum perspective is valuable if, for example, one wishes to design the optimal kernel to solve a given problem.
> * Our theorem regarding worse-than-chance generalization is new, surprising, and clearly illustrates the utility of Theorem 1.
> * The experiments of these works only study MSE, while we also study several other functionals of $\hat{f}$, giving a far more complete characterization of the properties of the function learned by the model.
> * Our derivation is quite different, even when it arrives at the same results. They use a lengthy but direct replica method calculation, while we use a much simpler method to find the functional form of learnability, then solve for $C$ using the no-free-lunch theorem. We feel that our derivation is, in many respects, easier to understand, and regardless it is quite interesting that two very different methods give the same result (both we and these authors find this intriguing).
> * Our observation and explanation of nonmonotonic loss curves at low $n$ is new. Actually, as far as we know, this is a qualitatively new phenomenon: we know of no other works, empirical or theoretical, that note that MSE can, e.g., be worse at one data point than at zero.
>
> You note that our Theorems 1 and 2 are quite simple. This is a virtue of a good theorem, not a drawback! *To the best of our knowledge, we have indeed proven two new, very simple facts about an old and widely-used ML method* (i.e. kernel regression). Though we understand this can be difficult to judge as a non-expert, we ask that our results be evaluated on their novelty and importance, not their complexity.
>
> Regarding our problem setting, we have updated the paper with a clearer motivation for the discrete domain and the missing discussion of the infinite-$M$ limit (thank you for noting this omission). The motivation is essentially that finite $M$ allows us to work with vectors and matrices instead of continuous operators and functions, which significantly simplifies our analysis relative to, e.g., Canatar et al. while allowing us to recover continuous results by later taking the infinite-$M$ limit. To take this limit, one essentially lets the discrete domain $\mathcal{X}$ grow to infinite size with points that densely approximate some desired measure in $\mathbb{R}^d$ (for example, points spread uniformly on the hypersphere). Random samples from $\mathcal{X}$ are thus essentially random samples from the desired measure. If this is not clear, we would be happy to explain further.
>
> Regarding the assumption that $m=1$, we agree that it was unclear and have clarified it. Here is the revised text:
> * Examining Equation 1, one finds that the $m$ indices of $f$ can each be treated separately: the learned $\hat{f}$ is equivalent to simply performing kernel regression with each of the $m$ indices as a scalar target function and then vectorizing the results. For simplicity, then, we hereafter assume $m=1$.
>
> To see this, it is sufficient to note that $\hat{f}(x) = A f(\mathcal{D})$, where $\hat{f}(x) \in \mathbb{R}^{1 \times m}$ is the learned function, $A \in \mathbb{R}^{1 \times n}$ is the product of two kernel matrices (see Eq. 1), and $f(x) \in \mathbb{R}^{n \times m}$ is the target function on the dataset. Because this is standard matrix multiplication, computing results for one column of $f(x)$ at a time gives the same result as computing the full matrix multiplication.
>
> As you suggest, we have rephrased our conclusion (and indeed our abstract and introduction) to clarify that our theoretical results treat only exact kernel regression, with the extension to finite nets purely empirical.
>
> Thank you again for your review. If you have any further critiques we can address or questions that would help you better assess the significance of our work, please let us know. We look forward to reading your response.

---

### Official Review · Reviewer_gb7t · 2021-11-09

**Correctness:** 2
**Technical Novelty And Significance:** 1
**Empirical Novelty And Significance:** 2
**Recommendation:** 3
**Confidence:** 5

**Main Review:**

Strength

This paper studies generalization of deep neural networks, a central topic of theoretical research in deep learning. This paper is written well and easy to follow.

Weakness

My major concern is that the key findings of this paper are either not novel which have been revealed in previous works, or lack comparison with existing works so it is unclear how significant the results are.

In particular, (1) the finding that target functions corresponding to fewer leading eigenfunctions of NTK are easier to learn have been revealed by the generalization bound in [a] (Theorem 5.1 of [a]). In addition, [b] shows that the empirical (training) loss drops fast when the target function lies on the subspace spanned by top-k eigenfunctions of the linear integral operator corresponding to the neural tangent kernel.

(2) The “no-free-lunch” theorem presented as Theorem 1 is based on the new notation of “learnability” defined in equation (2), and it would be important to connect this new notation to existing works so that one could understand the significance of the results. For example, the pessimistic result of Theorem 1 could be connected to the result in Section 3.2 of [c], where it was proved that there exists a difficult test data set which fails the trained ReLU neural network (the difficult test data and the training data can be generated from the same target function).

There are some other issues, for example, only asymptotic analysis (the width of the neural network goes to infinity) is presented. Note that all the referenced works [a-c] are based on neural networks with finite width, which is the case closer to the practice.


**Summary Of The Paper:**

This paper studies generalization capability of deep neural networks by the eigensystem of Neural Tangent Kernel (NTK). The findings of this paper include (1) target functions corresponding to fewer leading eigenfunctions of NTK are easier to learn (these target functions are learned faster); (2) a new “no-free-lunch” theorem characterizing a fundamental tradeoff in the inductive bias of wide neural networks: improving a network’s generalization for a given target function must worsen its generalization for orthogonal functions.

**Summary Of The Review:**

While this paper studies an important topic of deep learning, generalization capability of deep neural networks, the key findings of this paper are either not novel or lack comparison with existing works. I suggest the authors perform a thorough literature review in the recent progress of theoretical research in deep learning, and then carefully place their work under the context of the literature.

---

> ### Author Response · Authors · 2021-11-11
> **Request for references [a-c]**
>
> Thank you for your review. So that we might respond appropriately, could you please clarify which works you refer to with [a], [b], and [c]?

---

> > ### Comment · Reviewer_gb7t · 2021-11-11
> > **References**
> >
> > Please accept my apology for missing the key references, which are listed below.
> >
> > [a] Fine-Grained Analysis of Optimization and Generalization for Overparameterized Two-Layer Neural Networks, Arora et al. ICML 2019.
> >
> > [b] On Learning Over-parameterized Neural Networks: A Functional Approximation Perspective. Su et al. 2019.
> >
> > [c] Uniform convergence may be unable to explain generalization in deep learning. Nagarajan et al. NeurIPS 2019.

---

> > > ### Author Response · Authors · 2021-11-12
> > > **Response to reviewer comments and relation to [a-c]**
> > >
> > > We thank the reviewer for their comments and for suggesting the references [a-c], which we have read. We remain confident that our work is novel and significant, but agree that our paper should make more explicit its connections to and improvements over prior work. We will therefore add a dedicated "Relevant Background" section to the paper shortly in which refs. [a-c] will be discussed. We would first like to ensure that our positioning of our work relative to [a-c] indeed addresses the reviewer's comments, and so we discuss each of these references below.
> > >
> > > Our work differs from [a] as follows:
> > > * [a] only derives a *bound* for generalization error. As can be seen in Figure 2 of [a], this bound can be 6x higher than the true loss. Instead of a bound, we derive (accurate approximate) expressions for the generalization error itself - instead of merely an upper bound, our results directly provide quantities of interest. They are thus an advance over results of the type given in [a].
> > > * Figure 2 of [a] shows a case in which the bound happens to be proportional to the true L1 loss. We were curious as to whether this nice property still holds when the varied quantity is trainset size, so we tested their bound (with their network setting) on our hypersphere learning problem. [These](https://i.imgur.com/UlUW2oQ.png) are the results: triangles show true loss, while lines show the bound. Though [a]'s bound is indeed always an upper bound, it does not remain close to proportional to true loss. By contrast, our predictions give good agreement (qualitative and quantitative) with experiment (e.g. Figs. 1, 2), a clear improvement over [a].
> > > * Ref. [a] treats only a quite narrow setting: poly-wide 2-layer ReLU nets with the 2nd layer frozen and data on the hypersphere. By contrast, our theory is valid for *any* infinite-width network and should be expected to predict well whenever the net is wide enough to be in the NTK regime, which includes their poly-wide nets (which have width $\Omega(n^6)$ !). That said, we appreciate the distinction between infinite and large-but-finite.
> > > * Our works predict different quantities: the bound in [a] requires that the loss be 1-Lipshitz and thus applies to L1 loss but not MSE, while we predict MSE, "learnability," and indeed many other "error functions" of interest.
> > >
> > > If you feel it will improve understanding of our paper, we will add a short appendix including our above experiment and discussing these differences in detail.
> > >
> > > Ref. [b] chiefly differs from our work in that it studies training (i.e. dynamics of training loss), not generalization. Furthemore, they study the rate of eigenmode learning as a function of *training time*, not *number of samples*. In these regards, it is similar to [Cao et al](https://arxiv.org/abs/1912.01198). Though train loss dynamics are an important problem, our paper essentially "jumps" to the end of training, and so these works, though worth discussion, do not affect the novelty of our results.
> > >
> > > The "unlearnable function" in [c] is indeed worth discussing (though we note that they require the train and test sets to be mirror images of each other, while we make no such assumption). Indeed, such specialized "hard" functions are common in the literature (e.g. Fig. 7 of [Rahaman et al.](https://arxiv.org/pdf/1806.08734.pdf)). Our Thm. 2 states that there will *always* exist such pathologically hard functions for any input space, and it further gives a general recipe to find them via examination of the NTK. Our work is thus an important unification of prior works that exhibit special cases of hard functions. We note that several older works have studied certain functions that cannot be learned by a net without very high width, and so our work completes the picture by finding functions that are not learned *even by a net with infinite width.*
> > >
> > > Regarding our new notion of "learnability," it is best understood as the alignment of the learned function with the target function. Using the standard bias-variance decomposition of MSE, it is not hard to show that $\text{MSE} \ge (1 - \mathcal{L})^2$, so one cannot have low MSE without also having learnability near one. We will note this clearly when we push our revised paper.
> > >
> > > Lastly, we would like to address the criticism that we treat only infinite-width networks. General neural nets are so complex that a theorist must make some simplifying assumptions. Refs. [a-c] do indeed theoretically treat finite networks, but they require these nets have *two layers, ReLU activations, and very high width nonetheless* (and, in the case of [a], a frozen last layer and normalized data). By contrast, our theory *only* requires that the network has infinite width, and we instead rely on experiments to verify that our predictions show good agreement with finite networks. This is a different approach that allows us to address architectures that [a-c] could not.
> > >
> > > Thank you again for the review. We look forward to your response.

---

> > > > ### Comment · Reviewer_gb7t · 2021-11-28
> > > > **Lack of Novelty and Unclear Position of this Work in the Literature of Generalization Theory of Deep Neural Networks**
> > > >
> > > > I would like to thank the authors for the feedback to my review. However, my major concerns remain unaddressed after reading the authors' feedback. Therefore, I still recommend reject for this paper. Indeed, this work analyzes an unrealistic setting of neural network (infinite width), derives the generalization using a new metric of "learnability" which lacks discussion and comparison with prior work. More importantly, the key finding of the "learnability" metric, that is "A function is thus more learnable the more weight it has in high eigenvalue modes", is not novel and it was already revealed by previous works (such as [a]) based on a more realistic setting.
> > > >
> > > > (1) $\textbf {Lack of novelty regarding the key finding of this paper}$  As indicated in my initial review, the finding that target functions corresponding to fewer leading eigenfunctions of NTK are easier to learn has been revealed by the generalization bound in [a] (Theorem 5.1 of [a]). To further explain this point so that we can judge the novelty of this finding easier, please note that the generalization bound in [a] shows that the generalization error of a two-layer finite-width neural network is proportional to $\sqrt{\frac{\mathbf y^{\top} \mathbf H^{-1}  \mathbf y}{n}}$, where $\mathbf y$ is the ground truth label, and $\mathbf H$ is the neural tangent kernel. It can be seen that if the target function lies on the subspace spanned by the top-k eigenfunctions of the linear integral operator corresponding to the neural tangent kernel, then $\mathbf y$ lies on the space spanned by top-k eigenvectors of $\mathbf H$  (because $\mathbf y$ is generated by the target function), leading to a small value of the bound $\sqrt{\frac{\mathbf y^{\top} \mathbf H^{-1}  \mathbf y}{n}}$. [a] also shows empirical study with figures illustrating this finding.
> > > >
> > > > (2) As mentioned in (1), [a] and this work achieve the same finding. However, while [a] is based on finite-width neural network, this paper relies on the unrealistic setting of infinite-width neural network. Considering the difficulty of non-asymptotic analysis (such as finite-width) over asymptotic analysis (such as infinite width) in typical theoretical machine learning and deep learning problems, this work seems to reach to the same finding with an easier setting. This point can be seen by looking closer into the details of this work and [a]. While [a] makes efforts to specify the function class represented by two-layer neural network by fine-grained analysis of gradient descent starting from random initialization, this work is directly based on the learned function in equation [1] (so that the function class is an one-element class) without worrying about the complicated optimization process of deep neural networks. It should be noted that it is mostly the complicated optimization process that makes the generalization analysis of deep neural networks difficult.
> > > >
> > > > (3) Several arguments in the authors' feedback are surprisingly unreasonable. For example, the criticism "[a] only derives a bound for generalization error... Instead of a bound, we derive (accurate approximate) expressions for the generalization error itself". Note that this work does not derive the expression for the generalization error using the same metric as [a]! The metric used in this paper, the "learnability", has a significant problem in itself: it lacks comparison with prior work, so that it is hard to understand its significance and place this work in the literature of generalization theory of deep neural networks. Surprisingly, when arguing that the metric of "learnability" is useful in the feedback, the authors used a bound $\rm MSE \ge (1-\mathcal L)^2$, while they criticized that "[a] only derives a bound...".
> > > >
> > > > (4) Some argument is wrong in the authors' feedback. For example, it is not correct that "the bound in [a] requires that the loss be 1-Lipshitz and thus applies to L1 loss but not MSE". The 1-Lipshitzness used in [a] is only for simplicity of notations. The generalization error bound in [a] is based on Rademacher complexity, and the loss function $\ell$ can be trivially generalized to Lipshitz continuous function $\ell$ with arbitrary positive Lipshitz constant (including the MSE loss function) by the Talagrand's contraction principal of Rademacher averages.
> > > >
> > > > (5) As indicated in my initial review, the authors seem to be ignorant of many important works in theoretical deep leering, including but not limited to [a-c]. Indeed, there are many important works in generalization analysis of deep neural works which are not discussed in this work, such as papers referred by [a-c]. I suggest the authors make significantly more efforts in making new findings and rewrite this paper.

---

> > > > > ### Author Response · Authors · 2021-12-04
> > > > > **Response to reviewer response**
> > > > >
> > > > > We thank the reviewer for elaborating upon their critiques. Our high-level response is quite simple:
> > > > >
> > > > > * **We do not claim to be the first to show that "a function is more learnable the more weight it places in high eigenvalue modes"!** This is a well-established picture (as we discussed in our Introduction!), and if the reviewer believes that we claim this is a major novel finding, then we unfortunately fear they have significantly misunderstood our paper. We refer the reviewer to our Summary of Contributions, which consists of three long bullet points of novel and nuanced results, none of which make this claim and none of which appear in [a].
> > > > >
> > > > > * The reviewer has not noted any specific problems with our interpretation of [b,c] and their relationship to our work. We claim these are appropriate.
> > > > >
> > > > > * [a] is certainly an important paper, and we happily cite it, but the most relevant background works remain Bordelon et al. and the older GP literature.
> > > > >
> > > > > &nbsp;
> > > > >
> > > > > That said, we appreciate the reviewer's knowledge of [a], and since they took the time to enumerate critiques, we will respond to each.
> > > > >
> > > > > (1) See the first bullet point above: we do not claim this is a key finding!
> > > > >
> > > > > (2) Again, we do not "achieve the same finding" as [a]. Asymptotic analysis is indeed a price to pay, but with it we buy many new results (applicable to all wide architectures, not just that of [a]). We of course have great respect for work on wide-but-finite networks, but it is quite an extreme view to argue that work on infinite-width nets is not interesting!
> > > > >
> > > > > (3) Learnability is a new measure that, to our best knowledge, does not resemble anything in the literature beyond the bias term in the bias-variance decomposition. That said, we agree it could be motivated more clearly; the Introduction in the revision linked in our "Note to all reviewers" includes substantially more motivation for it. As for the bound, the reviewer appears to have misunderstood our point: *sure, MSE and learnability are related with a bound, but our paper provides non-bound expressions for both quantities.*
> > > > >
> > > > > (4) We thank the reviewer for the clarification. Given that the statement of [a]'s Thm. 3.3 begins with "For any 1-Lipschitz loss function...", we hope onlookers can understand why we assumed that the loss function had to be 1-Lipschitz.
> > > > >
> > > > > We thank the reviewer again for the review and extended discussion.

---

### Author Response · Authors · 2021-11-24
**Note to all reviewers**

We would like to thank each of you for your review and for your positive comments. As explained in individual responses, we believe we have understood and addressed each of your concerns. Aside from minor critiques, reviewer criticism essentially consisted of two points:

* The relationship of this paper to prior work was not sufficiently clear.
* Some of our statements regarding the implications of our results to finite neural networks were too strong.

We believe we have fully addressed these problems in the following ways:

* We have added dedicated Summary of Contributions and Related Works sections to our introduction. This includes Refs. [a,b] of Reviewer gb7t and clarifies how our work extends that of Bordelon et al and Canatar et al for Reviewers 7tiq and q2g8. We believe these place our paper in context and make it quite clear in what ways it is novel and significant.
* We rephrased much of the paper to make it clear that our results are derived for exact kernel regression, and the fact that they prove accurate for wide finite networks is merely a promising empirical observation.

We would also like to emphasize for the reviewers that our theoretical results apply generically to kernel regression with arbitrary kernels, not just NTKs, and their significance is thus commensurately greater. In particular, our observation of a simple conservation law governing the inductive bias of kernel regression (Theorem 1) is, to our knowledge, a novel and rather deep result regarding an extremely widely-used algorithm. We are thus puzzled by the comments of Reviewers gb7t and 7tiq that our results are not sufficiently novel, the criticism of Reviewer 7tiq that Theorem 1 is "simple," and the comment of Reviewer Mosm that our paper does not contain significant theory. We are genuinely open to the possibility that our results are indeed not significant in some manner we do not see, but if that is so, we would ask that these critiques be elaborated so that we can either respond or improve our paper. Alternatively, if the reviewers find their minds have changed, we would kindly ask that their scores change, too.

Lastly, we note that, in a revision that just missed the window, we have added more motivation to the introduction and changed the title. A PDF of this revision can be found at [this link](https://drive.google.com/file/d/1K3F-Gi0vmul6owOB26Sog28x0dtNVHAu/view?usp=sharing).

In summary, we believe we have fully addressed all specific criticisms raised by reviewers, and we would be happy to respond to further comments. Thank you all for your time.

---

### Decision · Program_Chairs · 2022-01-20

**Decision:**

Reject

**Comment:**

*Summary:* Study generalization in kernel regression discussing the NTK case and experiments on finite width nets.

*Strengths:*
- Mix of theoretical and empirical results in an important topic.
- Advances a promising recent line of work.

*Weaknesses:*
- Concerns about novelty and lack of comparison with existing works.
- Concerns about insufficient contextualization of new notion of learnability.
- Concerns about scope of results in relation to claims.

*Discussion:*

Reviewer gb7t (3) found their concerns about lack of novelty and comparison with prior works not sufficiently addressed in the authors’ responses. 7tiq (6) found the line of investigation promising, but also issues with presentation and found the theoretical results incremental. Mosm (5) finds that the theoretical part pertaining kernels does not offer much novelty and that the paper should have focused on the empirical study that links the NTK spectrum to generalization. q2g8 (8) confidently considers this a good paper. In their view it provides a nice theoretical analysis of generalization in the setting of kernel regression and the metric of learnability intuitive. However, they also found that the detests of the experiments are very artificial and problematic the desire of the article to extend the regime of the results to make claims about deep learning.

A the end of the discussion period, the official reviewer ratings are mixed 3,5,6,8, indicating various strengths and weaknesses (also in case of the most favorable reviews). From the reviews and discussion, I infer that the topic is worthwhile and relevant, but at the same time that the paper might not be sufficiently convincing in its current form. Therefore I lean to reject the paper. To arrive at a clear conclusion, I consulted two additional researchers.

*Additional assessment 1:*

The first additional assessment found the work 'underwhelming' but admitted there is a chance they might not have fully understood the work.

*Additional assessment 2:*

The second additional assessment provided following comments: The paper's first contribution (conservation law), I didn't see it elsewhere but I think it's quite expected. The testing performance of low-frequency target functions and high-frequency target functions are averaged. Thus the average performance is constant which is independent on the kernel. However, in practice, kernel learning performs well because real target functions have low frequency. And the very high-frequency functions are unrealistic.

About the paper's second contribution, I think the paper needs to explain how the result is different from Bordelon et al. (2020). I noticed that the method they use is different but the result seems quite similar. The paper also consider noiseless case and gives an approximation for MSE.

Also the paper should explain more about the approximation being used. For example, how much error the approximation introduce and how the approximation is different from Bordelon's approximation. I see that in the proof Φ is approximated by a matrix where each element is standard Gaussian. For me I can't understand why the approximation is reasonable.

I read through the reviews and rebuttals. I didn't see the discussion of the issue of approximation. But I think it's a major issue and the approximation is a very strong assumption. The appendix states: "we have made an approximation using the central limit theorem assuming that Φ is random with entries sampled i.i.d. from N (0, 1)". Here Φ is the matrix of eigenfunctions. Hence it is not clear how to apply the central limit theorem.

*Conclusion:*

I conclude that although the paper presents some interesting ideas on a relevant subject, it still has much room for improvement. Hence I recommend to reject this article. I encourage the authors to revise taking the above comments into consideration.